# Closed-loop Scaling Up for Visual Object Tracking

## Abstract

Thanks to the principles of the scaling law, current neural networks have experienced remarkable performance improvements. While much of the existing research has concentrated on upstream pretraining, the application of the scaling law to downstream vision tasks remains underexplored. Understanding the scaling law in downstream tasks can aid in the design of more effective models and training strategies. Thus, in this work, we aim to investigate the application of the scaling law to downstream vision tasks. Firstly, we explore the impact of three key factors of scaling law: training data volume, model size, and input resolution. We empirically verify that increasing each of these factors can lead to performance enhancements. Secondly, to address naive training's optimization challenges and lack of iterative refinement, we introduce DT-Training which leverages small teacher transfer and dual-branch alignment to further exploit model potential. Thirdly, building on DT-Training, we propose a closed-loop scaling strategy to incrementally scale the model step-by-step. Finally, our scaled model exhibits strong ability and outperforms existing counterparts across diverse test benchmarks. Extensive experiments also reveal the robust transfer ability of our model. Moreover, we validate the generalizability of the scaling law and our proposed DT-Training on other downstream vision tasks, reinforcing the broader applicability of our approach. We hope that our findings can deepen the understanding of the scaling law in downstream tasks and foster future developments on downstream tasks.

## 1 Introduction

The scaling law has demonstrated success and effectiveness across various domains, including speech (Radford et al., 2023), language (Brown, 2020; Devlin, 2018; Hoffmann et al., 2022; Raffel et al., 2020), vision (Kolesnikov et al., 2020; Zhai et al., 2022; Xie et al., 2023; Alabdulmohsin et al., 2024), and multi-modal (Pham et al., 2023; Jia et al., 2021; Alabdulmohsin et al., 2022; Radford et al., 2021; Ramesh et al., 2022; Rombach et al., 2022; Cherti et al., 2023). Training large models on extensive datasets over longer periods has consistently led to performance improvements and enhanced transfer ability. However, most of these efforts have concentrated on upstream pretraining stages. Although there have been a lot of works on scaling law training, these works mainly focus on the upstream pretraining. The application of scaling law principles to downstream vision tasks remains rarely explored. Understanding how scaling laws affect downstream vision tasks is crucial as it can inform the design of more effective models and training strategies.

In this work, we aim to explore the scaling law in downstream vision tasks. Recent researches (Kaplan et al., 2020; Brown, 2020) on scaling law in pretraining have prove that there exists a relationship between model performance and model parameters and size of dataset, which indicates that scaling up these factors can bring consistent performance improvement. Besides, larger input resolution of image can further result in enhanced accuracy (Zhai et al., 2022; Xie et al., 2023; Alabdulmohsin et al., 2024). Therefore, it is natural to ask whether downstream vision tasks possesses the same scaling signatures as the upstream tasks?

We take visual object tracking as case study to answer the above question. By systematically deflating model parameters, training data volume, and input image resolution, we investigate how these factors impact model performance in downstream vision tasks. As illustrated in Figure 1, our find-

ings reveal scaling patterns similar to those observed in upstream pretraining. Increasing model parameters, training data, and input resolution consistently results in stable accuracy enhancements.

Despite the improved accuracy, existing naive training methods encounter several issues based on our observation in Figure 1. Directly training a large model with extensive data may be difficult to optimize and challenging to fully harness its capabilities. Additionally, it is an open-loop training approach, failing to leverage knowledge gained from previous training. To address this, we introduce a novel training approach, DT-Training. In our DT-Training, a smaller model acts as a teacher, guiding the optimization of a larger model for smoother training. Additionally, DT-Training incorporates a dual-branch alignment technique, which applies random masks to input images and aligns outputs from both masked and unmasked images. This increases training difficulty, fully harnessing the model's potential. Building upon our DT-Training, we propose a closed-loop scaling up strategy. In this process, the small model from the previous iteration serves as a teacher, transferring knowledge to the larger model, which then becomes the foundation for the next iteration. This setup enables continuous iterative expansion, transforming the scaling process into an evolving cycle that consistently enhances performance.

Existing models often evaluate the performance on limited benchmarks that lack the diversity and complexity required to assess robustness in real-world scenarios. Thus, we introduce GTrack Bench, a comprehensive, challenging, and large-scale benchmark featuring 4,369 trajectories, approximately three times the size of existing benchmarks. With our DT-Training approach and closed-loop scaling strategy, our scaled model shows exceptional capabilities, outperforming current counterparts on GTrack Bench. Our model achieves 64.8 mean AUC, exceeding state-of-the-art methods by at least 1.4 mean AUC. Furthermore, it exhibits strong transferability, maintaining high performance even after compression and proving robust to multimodal data, such as depth maps. By integrating our model into the backbones of CompressTracker (Hong et al., 2024a) and OneTracker (Hong et al., 2024b), we achieve consistent performance improvements. Additionally, we also apply our strategy to other downstream vision tasks, such object detection, enhances the accuracy of Deformable DETR (Zhu et al., 2020) by 1.5 AP, which demonstrating the generalization ability of our method.

Our contribution can be summarized as following: (1) We take visual object tracking as case study to investigate scaling laws in downstream vision tasks, focusing on three key factors: model size, training data volume, and input resolution. Although increasing these factors can enhance performance, the improvement is often constrained by optimization challenges when training larger models. (2) We introduce a novel training approach DT-Training, which involves utilizing a smaller model to guide the training of a larger model, and aligning outputs from clean and masked images. Our DT-Training facilitates faster, smoother convergence and fully unlocks the model's potential. (3) We introduce a closed-loop scaling up strategy based on our DT-Training, transforming the scaling process into continuous, iterative optimization. This step-by-step evolution enables model to improve consistently across multiple iterations, fully harnessing its ability. (4) Our scaled model exhibits outstanding performance across various benchmarks and demonstrates robust transfer ability. Our model achieves 64.8 mean AUC on GTrack Bench, outperforming existing models by at least 1.4 mean AUC. Experiments on object detection demonstrates the generalization ability of our method.

## 2 SCALING LAW IN DOWNSTREAM VISION TASKS

In this section, we explore the impact of the three factors in downstream vision tasks: model size, training data, and image resolution, using visual object tracking as a case study. Our findings in the following can be applied to other tasks, such as object detection, too. We adopt OSTrack (Ye et al., 2022), which features a ViT (Dosovitskiy, 2020) encoder for joint feature extraction and temporal matching, and a lightweight decoder for box regression, for our experiments. This simple architecture allows us to effectively assess the impact of three factors in downstream vision tasks.

### 2.1 PIONEER EXPERIMENTS

To investigate the scaling laws affecting model performance, we systematically explore the effects of three key factors: model size, training data size, and input resolution, as shown in Figure 1. By keeping all other variables constant and scaling only one factor at a time, we observe a consistent pattern across all three dimensions: larger models, more extensive training data, and higher input

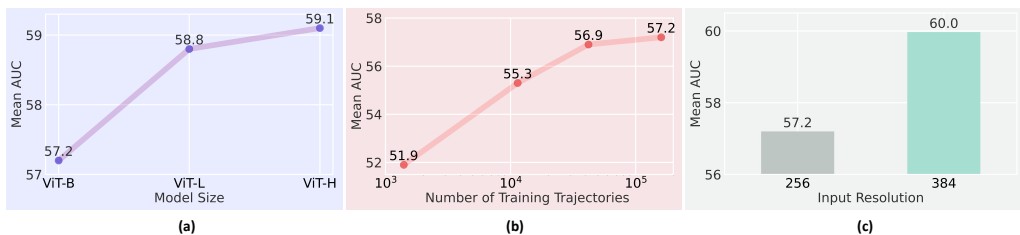

Figure 1: **Pioneer Experiments.** We investigate the impact of scaling law in downstream vision tasks: (a) model size, (b) training data, and (c) image resolution.

resolutions, each results in improved performance. These observations align with conclusions drawn from previous studies on scaling laws in pre-training tasks, highlighting the critical role of balancing model size, data quantity, and input resolution to optimize visual model performance.

## 2.2    Shortcuts of Naive Training

As shown in preceding pioneer experiments and Figure 1, we observe that while expanding certain factors like model size or training data can rapidly enhance model performance up to a specific threshold, beyond the certain point, further expansion results in less noticeable improvements. For example, a model using ViT-H as its backbone only achieves a $0.3\%$ increase in mean AUC compared to the ViT-L model. Similarly, the performance gains from expanding training data gradually slow down. We attribute these limitations to conventional training approaches. (1) **Convergence difficulty.** Firstly, training a large model directly on extensive datasets can be challenging to optimize due to the increased complexity and computational demands, often leading to issues like slow convergence or getting stuck in local minima. (2) **Underexplored Capabilities.** Traditional training often fails to fully exploit larger models' capabilities. While these models can capture stronger patterns, conventional training uses fixed training protocols and architectures may hinder their potential, resulting in suboptimal performance. (3) **Isolate optimization.** Besides, traditional methods follow a linear, open-loop process where each scaling step—whether increasing model size, data volume, or resolution—is treated in isolation. Models are trained independently, failing to utilize the insights and capabilities developed in previous training efforts. The absence of iterative knowledge-sharing process significantly limits the potential for more efficient optimization. This underscores the need for a new training approach to more effectively exploit model performance and a more integrated, close-loop approach to fully unlock the advantages of scaling laws.

## 3    Close-Loop Scaling Up Strategy

To address the aforementioned challenges, we introduce a novel training approach called DT-Training, and a closed-loop scaling up strategy. DT-Training integrates dual-branch alignment and small teacher transfer, to fully harness the potential of large models and improve performance. Moreover, DT-Training enables our closed-loop scaling up strategy. In this process, the small model from the previous iteration serves as a teacher to transfer knowledge to the larger model, which then becomes the starting point for the next iteration. This setup facilitates continuous iterative expansion, transforming the scaling process into an evolving cycle that consistently enhances performance.

### 3.1    DT-Training

While naive training can improve model performance by scaling up key factors in scaling laws, it faces significant limitations. Traditional training methods struggle to optimize large models effectively and fail to fully exploit their potential. To overcome these shortcuts, we introduce DT-Training as shown in Figure 2.

Directly training large models with excessive parameters often leads to challenges in pattern exploration and optimization difficulty. To solve the optimization difficulty problem, we introduce the small teacher transfer approach, where we employ a small pretrained model as a teacher to guide the optimization of the larger model, facilitating smoother learning and faster convergence for the larger model. Specifically, in our small teacher transfer, the original images $X$ are simultaneously fed into the training model $f$ and teacher model $\hat{f}$. To facilitate the optimization of the student model from

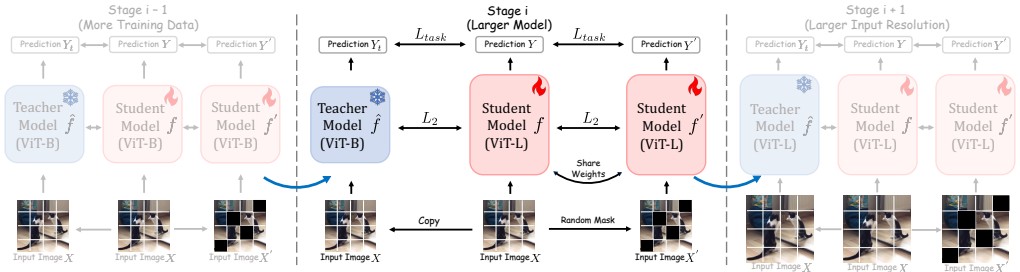

Figure 2: **Overview of our DT-Training and closed-loop scaling up strategy.** Our DT-Training includes small teacher transfer and dual-branch alignment. We provide an illustrative example of our closed-loop scaling up strategy to show a gradual increase in training data, model size, and image resolution. The order of expanding the three key factors is flexible and can be adjusted as needed.

different levels, we minimize the distances of both the prediction output and intermediate features. Given the output $Y$ and intermediate features $F$ obtained by the student model $(Y, F) = f(X)$ and teacher model $(\hat{Y}, \hat{F}) = \hat{f}(X)$, the objective function is formulated as:

$$L_{transfer}(f; \hat{f}) = L_{task}(Y, \hat{Y}) + L_2(F, \hat{F}), \tag{1}$$

where $L_2(F, \hat{F})$ denotes the L2 distance between the features $F$ and $\hat{F}$. $L_{task}(Y, \hat{Y})$ is utilized to calculate the difference between the outputs, which is task-specific. Note that we only update the parameters of the student model, and the teacher model is frozen. With Eq. (1), our method encourages comprehensive knowledge transfer between teacher and student models, facilitating smoother and more stable optimization for the student model.

To further exploit the ability of the model, we introduce the dual-branch alignment technique, where we apply random masks to input images and align the masked and unmasked image processes. By doing so, we improve the robustness of the model, thus unlocking the model's full potential. Specifically, to introduce additional complexity and promote generalization, we apply random masks to the origin image $X$, generating masked image $X^{'}$. This creates two parallel branches: a clean branch for the original image and a masked branch for the masked image, both of which share the same network weights. We then obtain the outputs and intermediate features of both the clean image $X$ and masked image $X^{'}$ by the shared student network $f$, formulated as:

$$(Y, F) = f(X), \quad (Y^{'}, F^{'}) = f(X^{'}), \tag{2}$$

where $Y^{'}, F^{'}$ are the predictions and intermediate features from the masked branch, respectively. To optimize the model, we first utilize use groundtruth supervision for the clean branch defined as:

$$L_{clean}(f) = L_{task}(Y, G), \tag{3}$$

where $L_{clean}$ denotes the task-specific loss for the clean branch and $G$ is the groundtruth label. Moreover, similar to Eq. 1, we align the clean and masked student branches by minimizing the distance between both the outputs and intermediate features. The loss for dual-branch alignment $L_{align}$ is then given by:

$$L_{align}(f) = L_{task}(Y, Y^{'}) + L_2(F, F^{'}). \tag{4}$$

While we use $L_{task}$ to compute the differences between the branches' outputs, more complex methods could also be applied. This loss function is designed to ensure both final predictions and intermediate features from the two branches are aligned, enhancing model's ability to generalize and leverage its full potential.

Finally, we combine the dual-branch alignment and small teacher transfer to jointly optimise the model. The overall loss function is formulated as:

$$L_{total}(f; \hat{f}) = L_{clean}(f) + \lambda_{transfer} L_{transfer}(f; \hat{f}) + \lambda_{align} L_{align}(f), \tag{5}$$

where $\lambda_{align}$ and $\lambda_{transfer}$ serve as the regularization parameters to balance these components. Overall, the knowledge transfer from the teacher to the student model allows the student to leverage the teacher's pretrained understanding of the task, enabling faster convergence and more efficient learning. Additionally, the masked branch operates with incomplete visual information due to occlusions caused by the random masks. This missing local information makes the task more demanding for the masked branch compared to the clean branch. Aligning the two branches enhances the

Table 1: **GTrack Bench statics.** GTrack Bench consists of 12 challenging benchmarks and roughly 4 times the trajectory number provided by current popular benchmarks.

|  | LaSOT | LaSOT$_{ext}$ | TrackingNet | TNL2K | UAV123 | Avist | LaGOT | LaTOT | HOOT | VideoCube | MOSE | OVIS | Sum |
|---|---|---|---|---|---|---|---|---|---|---|---|---|---|
| Trajectories | 280 | 150 | 511 | 600 | 123 | 120 | 850 | 165 | 130 | 50 | 531 | 859 | 4369 |
| Videos | 280 | 150 | 511 | 600 | 123 | 120 | 280 | 165 | 130 | 50 | 200 | 200 | 3379 |
| Mean Frames | 2512 | 2395 | 441 | 697 | 1247 | 666 | 2512 | 684 | 730 | 14267 | 70 | 78 | - |

robustness of the student model to incomplete and noisy data, resulting in stronger representational capabilities. Through the combination of dual-branch alignment and teacher model transfer, we address the optimization difficulty of naive training approaches and further exploit model's capability.

## 3.2 CLOSED-LOOP SCALING UP

To solve the isolate optimization problem, we further propose the closed-loop scaling up strategy built on the DT-Training by introducing a feedback mechanism to enable continuous, iterative optimization throughout the scaling process. As shown in Figure 2, our closed-loop strategy progressively expands any key factor of scaling laws: model size, data size, and input resolution, which we explore in Section 2.

Given its iterative nature, each training phase can be viewed as a stage with different data volume $\gamma$, model parameters $\theta$, and input resolution $\mu$. These factors scale up as the iteration process increases. Based on the key idea of using a smaller teacher model to guide larger student one in DT-Training, we use the trained student model $f_{i-1}$ as the teacher model for stage $i$. The larger scale student model is denoted as $f_i$. We use the same training objective functions within the DT-Training framework for each iteration, which includes dual-branch alignment and small teacher transfer. The goal of each iteration is to incrementally scale the student model and enhance its performance by leveraging the knowledge embedded in the teacher model.

At the start of the $i$th iteration, the model from the previous iteration, $f_{i-1}$, though smaller or less accurate, contains valuable knowledge that has been optimized on the tasks encountered in earlier iteration. This model serves as the teacher in the DT-Training process, facilitating faster convergence and smoother optimization for the current iteration. In each iteration, one or more of the three scaling factors is increased, allowing the model to progressively evolve and improve. The optimization function for the $i$th iteration can be formulated as:

$$L_{total}(f_i; f_{i-1}|\theta_i, \gamma_i, \mu_i), \tag{6}$$

where $\theta_i$, $\gamma_i$, and $\mu_i$ denote parameter amounts, data volume, and input resolution in stage $i$, respectively. After the $i$th stage completes, the model $f_i$ becomes the new teacher for the subsequent $i + 1$th stage, continuing the cycle of iterative scaling and improvement. In each new iteration stage, we scale the model by either increasing its capacity, expanding the dataset size, or enhancing the input resolution. This ensures that the student model is progressively larger and more capable while leveraging the knowledge acquired in previous iterations. By iteratively expanding these key factors and continuously transferring knowledge between models, our closed-loop scaling strategy guarantees that each iteration benefits from prior learning. This approach ultimately leads to more robust and efficient scaling across model size, data, and resolution, enhancing overall performance.

Our DT-Training enables the feasibility of a closed-loop scaling strategy, offering key advantages over traditional methods. First, the iterative teacher-student relationship allows each new student model to inherit the accumulated knowledge of previous iterations, leading to faster convergence and better generalization. Second, while conventional training often faces diminishing returns as models are scaled, our strategy transforms scaling into an iterative refinement process, ensuring consistent improvement. Additionally, the closed-loop scaling strategy offers excellent scalability, making it suitable for progressively larger models and more complex datasets as the training advances.

## 4 EXPERIMENTS

### 4.1 IMPLEMENT DETAILS

Our DT-Training approach and closed-loop scaling up strategy are general and can be applied to any kind of downstream vision models. Because we take visual object tracking as a case study, we select

Table 2: **Effectiveness of DT-Training.** We compare the performance between our DT-Training and the conventional training approach under the same conditions. For 'Baseline-B-256-N', 'Baseline' indicates model name, 'B' refers to ViT-B, '256' specifies the input resolution, and 'N' represents training data. N refers to normally used four tracking datasets, and M represents more training data.

| Model | LaSOT | | | LaSOT$_{ext}$ | | | TNL2K | | | Mean |
|---|---|---|---|---|---|---|---|---|---|---|
| | AUC | P$_{Norm}$ | P | AUC | P$_{Norm}$ | P | AUC | P$_{Norm}$ | P | AUC |
| Baseline-B-256-N | 68.4 | 77.8 | 74.2 | 47.0 | 57.0 | 52.9 | 56.4 | 71.7 | 58.4 | 57.3 |
| *Training Data Scale Up* | | | | | | | | | | |
| Baseline-B-256-M | 68.6 | 78.3 | 74.2 | 47.3 | 55.9 | 51.8 | 60.5 | 76.9 | 65.0 | 58.8 |
| **Ours-B-256-M** | **69.5** | **79.2** | **75.3** | **47.9** | **57.5** | **53.5** | **61.2** | **77.2** | **65.0** | **59.5** |
| *Model Size Scale Up* | | | | | | | | | | |
| Baseline-L-256-N | 70.0 | 79.2 | 76.3 | 46.6 | 56.9 | 53.0 | 59.6 | 71.9 | 58.9 | 58.7 |
| **Ours-L-256-N** | **71.0** | **80.9** | **77.2** | **46.0** | **55.9** | **52.2** | **60.1** | **72.6** | **59.5** | **59.2** |
| *Input Resolution Scale Up* | | | | | | | | | | |
| Baseline-B-384-N | 70.0 | 79.4 | 76.1 | 51.4 | 62.2 | 58.1 | 58.5 | 70.7 | 57.0 | 60.0 |
| **Ours-B-384-N** | **70.6** | **80.3** | **76.8** | **51.9** | **62.6** | **58.6** | **59.4** | **72.0** | **58.1** | **60.6** |

OSTrack (Ye et al., 2022) as baseline due to its simplicity and effectiveness. The training datasets include LaSOT (Fan et al., 2019), TrackingNet (Muller et al., 2018), GOT-10K (Huang et al., 2019), and COCO (Lin et al., 2014), aligning with OSTrack (Ye et al., 2022) and MixFormerV2 (Cui et al., 2024). However, these datasets alone do not provide sufficient data to fully train a highly capable tracking model, so we convert datasets from related tasks, such as multi-object tracking, video object segmentation, and open-world object tracking and segmentation, into a single object tracking format. Each video in these additional datasets may contain multiple trajectories, as opposed to only one labeled object's trajectory in visual object tracking. By incorporating a significant number of training trajectories, we effectively expand our training data to four times its original size, surpassing what was available in the initial four datasets. See Appendix A.3 for more details about training data.

We train the model with AdamW optimizer (Loshchilov & Hutter, 2017), with a weight decay of $10^{-4}$ and an initial learning rate of $4 \times 10^{-4}$. The total training epochs is 300 with 60K image pairs per epoch and the learning rate is reduced by a factor of 10 after 240 epochs. We employ a batch size of 256. The search and template images are resized to resolutions of $256 \times 256$ and $128 \times 128$ resolutions, respectively. We set $\lambda_{align}$ as 0.1. $\lambda_{transfer}$ are set as 0.5 for the first 270 epochs and reduc to 0.0 for the last 30 epochs. The mask ratio is gradually increased from 0.05 to 0.4. We initialize the model with the pretrained parameters from MAE. To maximize the benefit of extensive training data, we employ a balanced sampling strategy to ensure that larger datasets do not overshadow smaller ones.

## 4.2 GTRACK BENCH

Existing tracking models (Cui et al., 2022; 2024; Ye et al., 2022; Bai et al., 2023) tend to assess performance on a limited number of benchmarks (about 3-4, covering approximately 1000 trajectories), including TrackingNet (Muller et al., 2018), GOT-10K (Huang et al., 2019), and LaSOT (Fan et al., 2019). However, these datasets offer insufficient diversity, and the videos lack the complexity required to assess model robustness in real-world scenarios. Thus, we introduce a comprehensive and challenging benchmark, called General Track Bench (GTrack Bench), designed to comprehensively evaluate the ability of tracking models in diverse scenes. GTrack Bench consists of 3379 videos from 12 datasets, with a total of 4369 trajectories, roughly 3 times the number provided by current popular benchmarks (around 1000 trajectories). The statistics of these 12 datasets and GTrack Bench are summarized in Table 1. The collection includes 10 tracking datasets, along with one video object segmentation (VOS) dataset and one video instance segmentation (VIS) dataset. The 10 tracking datasets not only include some of commonly used datasets, such as TrackingNet, La-SOT, LaSOT$_{ext}$, and UAV123 (Mueller et al., 2016), as well as more challenging, recently proposed datasets tailored to complex scenarios, e.g. TNL2K (Wang et al., 2021c), and Avist (Noman et al., 2022). In addition to standard tracking datasets, we incorporate benchmarks MOSE (Ding et al., 2023) and OVIS (Qi et al., 2022) from VOS and VIS tasks and convert them into tracking format. These datasets capture complex scenes where target objects frequently experience occlusions, presenting a higher degree of difficulty. We calculate the mean results of each benchmark to serve as the

Table 3: **Effectiveness of closed-loop scaling up strategy.** We compare the performance of our closed-loop scaling up strategy with naive training on GTrack Bench.

| Model | LaSOT | LaSOT$_{ext}$ | TrackingNet | TNL2K | UAV123 | Avist | LaGOT | LaTOT | HOOT | VideoCube | MOSE | OVIS | Mean |
|---|---|---|---|---|---|---|---|---|---|---|---|---|---|
| Baseline-B-256-N | 68.4 | 47.0 | 83.5 | 56.4 | 67.8 | 57.0 | 61.9 | 28.9 | 56.4 | 45.5 | 51.4 | 55.3 | 59.4 |
| Ours-B-256-M | **69.5** | **47.9** | **83.6** | **61.2** | **69.2** | **57.6** | **63.1** | **30.6** | **56.5** | **47.4** | **55.5** | **60.1** | **62.0** |
| Baseline-L-256-N | 70.0 | 46.6 | **84.4** | 59.6 | 67.9 | 58.3 | 62.4 | 30.2 | 61.1 | 47.4 | 52.4 | 57.5 | 60.9 |
| Ours-L-256-M | **71.6** | **48.2** | 84.2 | **65.0** | **69.1** | **60.1** | **65.2** | **30.5** | **62.0** | **48.5** | **55.6** | **61.2** | **63.6** |
| Baseline-L-384-N | 70.8 | 47.0 | **85.0** | 60.5 | **70.3** | 59.6 | 63.4 | 31.0 | 61.8 | 48.6 | **57.5** | **63.3** | 63.4 |
| Ours-L-384-M | **73.1** | **53.0** | 84.7 | **66.3** | 69.7 | **60.5** | **67.3** | **32.0** | **62.0** | **53.1** | 55.7 | 61.5 | **64.8** |

Table 4: **Comparison with state-of-the-art models on GTrack Bench.** Our models significantly outperform state-of-the-art counterparts, highlighting the effectiveness of our DT-Training and closed-loop scaling up strategy.

| Model | LaSOT | LaSOT$_{ext}$ | TrackingNet | TNL2K | UAV123 | Avist | LaGOT | LaTOT | HOOT | VideoCube | MOSE | OVIS | Mean |
|---|---|---|---|---|---|---|---|---|---|---|---|---|---|
| Baseline-B-256-N | 68.4 | 47.0 | 83.5 | 55.9 | 70.7 | 57.0 | 61.9 | 28.9 | 56.4 | 45.5 | 51.4 | 55.3 | 59.4 |
| GRM-Base | 69.9 | 47.3 | 84.0 | 57.0 | 70.2 | 54.5 | 62.4 | 28.8 | 56.7 | 45.4 | 52.4 | 56.7 | 60.2 |
| SeqTrack-Base | 69.9 | 49.5 | 83.3 | 54.9 | 69.2 | 56.8 | 63.5 | 29.8 | 50.3 | 48.5 | 49.8 | 54.7 | 59.3 |
| ARTrack-Base | 70.4 | 46.4 | 84.2 | 57.5 | 67.7 | 59.9 | 62.7 | 30.8 | 56.2 | 44.4 | 52.4 | 57.7 | 60.6 |
| ARTrackV2-Base | 71.6 | 50.8 | 84.9 | 59.2 | 69.9 | - | - | - | - | - | - | - | - |
| **Ours-B-256-M** | 69.5 | 47.9 | 83.6 | **61.2** | 69.2 | 57.6 | 63.1 | 30.6 | 56.5 | 47.4 | 55.5 | 60.1 | 62.0 |
| Baseline-L-256-N | 69.9 | 47.1 | 84.4 | 59.6 | 67.9 | 58.3 | 62.4 | 30.2 | 61.1 | 47.4 | 52.4 | 57.5 | 60.9 |
| SeqTrack-L | 72.1 | 50.5 | 85.0 | 56.9 | 69.7 | 61.1 | 65.5 | 31.5 | 51.4 | 51.2 | 52.8 | 58.2 | 61.7 |
| **Ours-L-256-M** | 71.6 | 48.2 | 84.2 | **65.0** | 69.1 | 60.1 | 65.2 | 30.5 | 62.0 | 48.5 | 55.6 | 61.2 | 63.6 |
| Baseline-L-384-N | 70.8 | 47.0 | 85.0 | 60.5 | 70.3 | 59.6 | 63.4 | 31.0 | 61.8 | 48.6 | 57.5 | 63.3 | 63.4 |
| GRM-L320 | 71.4 | 51.5 | 84.4 | 58.2 | 70.8 | 57.5 | 64.8 | 32.5 | 58.5 | 50.9 | 51.5 | 56.6 | 61.3 |
| SeqTrack-L384 | 72.5 | 50.7 | 85.5 | 57.8 | 68.5 | 63.1 | 65.6 | 30.8 | 53.2 | 51.8 | 54.3 | 59.8 | 62.4 |
| ARTrack-L384 | 73.1 | 52.4 | 85.6 | 61.1 | 69.2 | 64.5 | 66.2 | 34.2 | 63.1 | 43.0 | 55.3 | 61.3 | 63.9 |
| ARTrackV2-L384 | **73.6** | **53.4** | **86.1** | 61.6 | 71.7 | - | - | - | - | - | - | - | - |
| **Ours-L-384-M** | 73.1 | 53.0 | 84.7 | **66.3** | 69.7 | 60.5 | 67.3 | 32.0 | 62.0 | 53.1 | 55.7 | 61.5 | **64.8** |

final score. By integrating this diverse range of datasets, GTrack Bench provides a comprehensive and realistic framework for evaluating model performance across varied and challenging environments. This enhanced benchmark allows for a more robust assessment of tracking models' abilities in real-world scenarios, and we will use GTrack Bench for evaluation in the following experiments. Please see Appendix A.2 for more details about our GTrack Bench.

## 4.3 CLOSE-LOOP SCALING UP

To validate the effectiveness of our DT-Training and close-loop scaling strategy, we conducted a comparison between models trained using our approach and those trained with a traditional, naive training method.

**Effectiveness and Generalization of DT-Training.** Firstly, to assess the generalization capability and effectiveness of our DT-Training method, we start with a baseline model trained on a limited set of commonly used datasets (*e.g.* COCO (Lin et al., 2014), TrackingNet (Muller et al., 2018), LaSOT (Fan et al., 2019), and GOT-10k (Huang et al., 2019)), following previous works (Ye et al., 2022; Bai et al., 2024; Cui et al., 2022). We then independently examine the impact of three critical factors in scaling law: model size, training data, and image resolution, as explored in Section 2. The results, presented in Table 2, demonstrate that our DT-Training consistently surpasses traditional training approaches across the three scaling conditions. Specifically, when only the training data was scaled up, we expand the dataset beyond the initial set (*e.g.*, COCO, TrackingNet, LaSOT, GOT-10k) by adding more diverse and larger-scale datasets, which results in a 0.7% increase in the mean AUC score across three datasets compared to naive training. In cases where only the model size is scaled up, we increase the complexity of the model by using a larger architecture, moving from ViT-B to ViT-L. This adjustment yields a 0.5% increase in the mean AUC score over naive training. Additionally, when the image resolution is increased from 256 to 384, we observe a performance boost of approximately 0.6% in mean accuracy. In summary, our DT-Training demonstrates significant effectiveness, as evidenced by consistent performance improvements across the three scaling conditions compared to traditional training methods.

**Effectiveness of close-loop scaling up strategy.** We conduct experiments to evaluate the effectiveness of our close-loop scaling up strategy. We also adopt the baseline model trained on the four limited datasets (*e.g.*, COCO, TrackingNet, LaSOT, GOT-10k) to serve as the start point of our close-loop scaling up process. We then progressively expand the training data, the model size, and the resolution of the input images by levaraging our DT-Training. Besides, we finetune the scaled model on LaSOT for 40 epochs. We compare the result with naive training the baseline model on the four limited datasets by using the GTrack Bench and show the result in Table 3. We record the

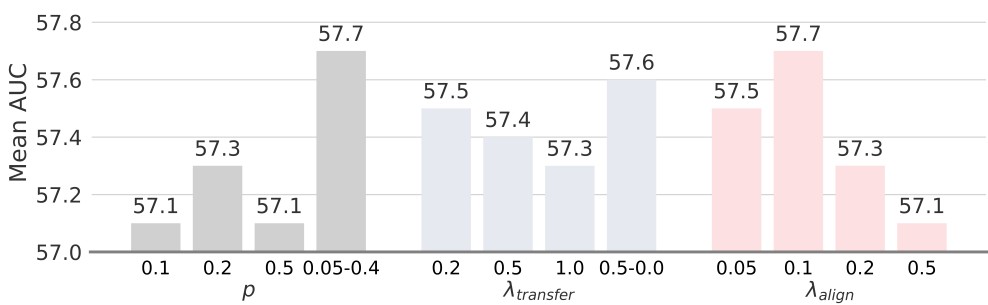

Figure 3: **Ablation study on mask ration and regularization parameters.** We conduct experiments to explore the impact of mask ration $p$ and regularization parameters $\lambda_{transfer}$ and $\lambda_{align}$.

AUC score of each benchmark and the mean score. Our model share the same inference speed with baseline model. Our model has a performance gain of at least $2\%$ in the average AUC over ten benchmarks over normal training in all different settings. Our training manner not only is proven to be effective when scaling a single element, but also demonstrate strong effectiveness and flexible scalability in closed-loop scaling experiments. This also demonstrates the superiority of our training manner and close-loop scaling up strategy compared to naive training.

**Comparison with existing models.** To further verify the effectiveness of our closed-loop scaling up strategy, we compare our models with state-of-the-art counterparts on GTrack Bench, as presented in Table 4. Our models achieve competitive accuracy, surpassing existing models by at least 1.4 mean AUC. Notably, while existing models such as ARTrack (Bai et al., 2024), and SeqTrack (Chen et al., 2023) rely on complex architectural designs for performance gains, our models obtain superior results with a simpler structure. This underscores the effectiveness of our DT-Training and closed-loop scaling strategy.

### 4.4 ABLATION STUDY

To verify the effectiveness of our proposed DT-Training, we conduct a comprehensive analysis of its various components, performing detailed exploratory studies. Unless otherwise noted, Unless otherwise specified, the following experiments use a ViT-B model trained on four datasets (COCO, TrackingNet, LaSOT, and GOT-10k) as a teacher model to train another ViT-B tracker on the same datasets, for the purpose of eliminating the influence of other factors, such as resolution, training data volume, and model parameter size.

#### 4.4.1 SMALL TEACHER TRANSFER & MASK ALIGNMENT.

We conduct experiments to investigate the effects of teacher transfer and mask alignment, with the results presented in Table 5. It can be observed that both the small teacher transfer (# 2) and mask alignment (# 3) can enhance accuracy compared to naive training (# 1). Moreover, combining small teacher transfer with mask alignment (# 4) can further improve model performance. Importantly, by using the same training data, model size, and input image resolution as the baseline training (# 1), our approach significantly boosts performance, highlighting effectiveness of our DT-Training.

Table 5: **Ablation Study on Small Teacher Transfer & Mask Alignment.** We investigate the effects of teacher transfer and mask alignment.

| # | Teacher | Mask | LaSOT | LaSOT$_{ext}$ | TNL2K | Mean |
|---|---------|------|-------|---------------|-------|------|
| 1 | | | 68.4 | 47.0 | 56.4 | 57.3 |
| 2 | ✓ | | 68.9 | 47.1 | 56.7 | 57.6 |
| 3 | | ✓ | 69.4 | 47.2 | 56.5 | 57.7 |
| 4 | ✓ | ✓ | 70.1 | 47.4 | 56.6 | 58.0 |

#### 4.4.2 MASK RATIO.

To explore the influence of mask ratio $p$ on mask alignment, we test model performance across different mask ratio and record results on the left side of Figure 3. The results reveal that a low mask ratio (0.1 and 0.2) fails to fully exploit the model's capabilities, while an excessively high mask ratio (0.5) increases training difficulty, negatively impacting performance. Thus, selecting an appropriate mask ratio is crucial to maximizing performance. We begin with a lower mask ratio to allow for faster learning and, as training stabilizes, gradually increase the mask ratio to enhance difficulty,

Table 6: **Compression experiments.** Our model maintains competitive accuracy after compression.

| Method | LaSOT | | | LaSOT$_{ext}$ | | TNL2K | | TrackingNet | | | UAV123 | |
|---|---|---|---|---|---|---|---|---|---|---|---|---|
| | AUC | P$_{Norm}$ | P | AUC | P | AUC | P | AUC | P$_{Norm}$ | P | AUC | P |
| HiT-Base (Kang et al., 2023) | 64.6 | 73.3 | 68.1 | 44.1 | - | - | - | 80.0 | 84.4 | 77.3 | 65.6 | - |
| HiT-Samll (Kang et al., 2023) | 60.5 | 68.3 | 61.5 | 40.4 | - | - | - | 77.7 | 81.9 | 73.1 | 63.3 | - |
| HiT-Tiny (Kang et al., 2023) | 54.8 | 60.5 | 52.9 | 35.8 | - | - | - | 74.6 | 78.1 | 68.8 | 53.2 | - |
| SMAT (Gopal & Amer, 2024) | 61.7 | 71.1 | 64.6 | - | - | - | - | 78.6 | 84.2 | 75.6 | 64.3 | 83.9 |
| MixFormerV2-S (Cui et al., 2024) | 60.6 | 69.9 | 60.4 | 43.6 | 46.2 | 48.3 | 43.0 | 75.8 | 81.1 | 70.4 | 65.8 | 86.8 |
| CompressTracker-4 (Hong et al., 2024a) | 66.1 | 75.2 | 70.6 | 45.7 | 50.8 | 53.6 | 52.5 | 82.1 | 87.6 | 80.1 | 67.4 | 88.0 |
| **CompressTracker-4-Ours** | 66.9 | 76.3 | 71.7 | 46.0 | 51.4 | 54.8 | 54.9 | 82.6 | 87.9 | 80.5 | 67.9 | 88.3 |

Table 7: **Multi-modal robustness experiments.** Our model is robust to multi-modal data.

| | | **RGB+D Tracking** | | | | | | |
|---|---|---|---|---|---|---|---|---|
| | | DeT | OSTrack | SPT | ProTrack | ViPT | OneTracker | **OneTracker** |
| | | Yan et al. (2021b) | Ye et al. (2022) | Zhu et al. (2022) | Yang et al. (2022) | Zhu et al. (2023a) | Hong et al. (2024b) | **Ours** |
| DepthTrack | F-score(↑) | 53.2 | 52.9 | 53.8 | 57.8 | 59.4 | 60.9 | 61.6 |
| Yan et al. (2021c) | R(↑) | 50.6 | 52.2 | 54.9 | 57.3 | 59.6 | 60.4 | 61.2 |
| | P(↑) | 56.0 | 53.6 | 52.7 | 58.3 | 59.2 | 60.7 | 61.5 |
| VOT | EAO(↑) | 65.7 | 67.6 | 65.1 | 65.1 | 72.1 | 72.7 | 73.5 |
| RGBD2022 | Accuracy(↑) | 76.0 | 80.3 | 79.8 | 80.1 | 81.5 | 81.9 | 83.0 |
| Kristan et al. (2023) | Robustness(↑) | 84.5 | 83.3 | 85.1 | 80.2 | 87.1 | 87.2 | 88.1 |
| | | **RGB+T Tracking** | | | | | | |
| | | APFNet | OSTrack | TransT | ProTrack | ViPT | OneTracker | **OneTracker** |
| | | Xiao et al. (2022) | Ye et al. (2022) | Chen et al. (2021) | Yang et al. (2022) | Zhu et al. (2023a) | Hong et al. (2024b) | **Ours** |
| LasHeR | PR(↑) | 50.0 | 51.5 | 52.4 | 53.8 | 65.1 | 67.2 | 68.3 |
| Li et al. (2021) | SR(↑) | 36.2 | 39.4 | 41.2 | 42.0 | 52.5 | 53.8 | 55.1 |
| RGBT234 | MPR(↑) | 79.0 | 82.3 | 82.7 | 79.5 | 83.5 | 85.7 | 86.2 |
| Li et al. (2019b) | MSR(↑) | 57.3 | 57.5 | 57.9 | 59.9 | 61.7 | 64.2 | 64.8 |
| | | **RGB+E Tracking** | | | | | | |
| | | LTMU | SiamRCNN | MDNet | OSTrack | ViPT | OneTracker | **OneTracker** |
| | | Dai et al. (2020) | Voigtlaender et al. (2020) | Nam & Han (2016) | Ye et al. (2022) | Zhu et al. (2023a) | Hong et al. (2024b) | **Ours** |
| VisEvent | MPR(↑) | 65.5 | 65.9 | 66.1 | 69.5 | 75.8 | 76.7 | 77.4 |
| Wang et al. (2021b) | MSR(↑) | 45.9 | 49.9 | - | 53.4 | 59.2 | 60.8 | 61.7 |

thereby fully harnessing the model's potential (0.05-0.4). This adaptive strategy ensures the model achieves optimal performance by balancing learning ease and challenge.

### 4.4.3 REGULARIZATION PARAMETERS.

The regularization parameters also have influence on model performance. As shown in the middle of Figure 3, small teacher transfer enhances model performance, but different $\lambda_{transfer}$ exert a relatively minor influence. In the fourth bar, teacher transfer is employed during the initial 270 epochs to boost training efficiency and performance. In the final 30 epochs, teacher transfer is disabled, allowing the model to independently refine its capabilities, thereby further enhancing performance. This method effectively capitalizes on the strengths of teacher transfer while enabling autonomous learning, resulting in superior model performance. In the right side of Figure 3, we examine the impact of $\lambda_{align}$. We find that both overly high and low $\lambda_{align}$ can negatively impair effectiveness, highlighting the importance of selecting an appropriate $\lambda_{align}$ for optimal results.

## 5 TRANSFER ABILITY PROBING

In the previous section, we validate the effectiveness of our proposed closed-loop scaling up strategy, but the transfer ability of our model has not been verified. While our model demonstrates excellent performance across numerous datasets, the transfer ability remains unexplored. Therefore, in this section, we conduct additional experiments to thoroughly evaluate the model's transfer capabilities.

**Model Compression.** Firstly, we aim to verify whether our model can maintain its excellent performance after compression. We follow CompressTracker (Hong et al., 2024a) framework and compress our scaled ViT-B model into a smaller version with just four transformer layers. Except for using a different initial teacher model, all other training parameters, such as data and epochs, remain consistent. As shown in Table 6, our model achieves superior performance, recording a 66.9% AUC on LaSOT benchmarks, which is a 0.8% AUC improvement over the original CompressTracker., thanks to our stronger model. Additionally, our model outperforms other lightweight tracking models, confirming its ability to maintain excellent performance after compression.

**Robustness to multi-modal data.** Furthermore, we investigate the the generalization ability of our model on multimodal data such as thermal maps. By adopting the OneTracker (Hong et al., 2024b) architecture, we explore the adaptability of our models to different modalities, including depth, thermal, and event maps. As shown in Table 7, our model shows strong generalization to multi-

modal data. Through replacing the backbone of OneTracker (Hong et al., 2024b) with our model, OneTracker obtains consistent performance improvement across various multimodal benchmarks. These findings, with our previous experiments, underscore robust transferability of our model.

## 5.1 GENERALIZATION EXPERIMENTS

Our DT-Training and closed-loop scaling up strategy can be applied to other downstream vision tasks. To verify the generalization capability of our method, we conduct experiments on object detection. We apply our method to Deformable DETR (Zhu et al., 2020) and train it on COCO (Lin et al., 2014) dataset for 50

Table 8: **Generalization Experiments.** Our DT-Training can also be applied to other tasks, such as object detection.

| Model | AP | $AP_S$ | $AP_M$ | $AP_L$ |
|---|---|---|---|---|
| Deformable DETR-R50 | 44.5 | 27.1 | 47.6 | 59.6 |
| **Deformable DETR-R50-Ours** | **46.0** | **27.4** | **49.3** | **61.1** |

epochs, maintaining the original settings. As show in Table 8, our method yields a 1.5 AP performance improvement over origin Deformable DETR under identical settings. Experiments on both tracking and object detection demonstrate that our model effectively operates on both CNN networks and Transformer architectures, demonstrating generalization ability of our method.

## 6 RELATED WORKS

### 6.1 SCALING LAW IN UPSTREAM TASKS

Scaling laws in neural language processing and vision pretraining tasks have been extensively studied in prior works (Hestness et al., 2017; Sun et al., 2017; Brown, 2020). Studies such as (Hoffmann et al., 2022; Kaplan et al., 2020; Tay et al., 2021; Touvron et al., 2023) explore neural scaling laws in language models, demonstrating a power law relationship between model performance and the scale of model size, data, and training compute. Similar power law dependencies have also been observed in vision tasks (Riquelme et al., 2021; Zhai et al., 2022; Dehghani et al., 2023; Kolesnikov et al., 2020; Xie et al., 2023; Alabdulmohsin et al., 2024). Additionally, works like (Radford et al., 2021; Pham et al., 2023; Jia et al., 2021; Alabdulmohsin et al., 2022; Radford et al., 2021; Ramesh et al., 2022; Rombach et al., 2022; Cherti et al., 2023; Fang et al., 2022; Yu et al., 2022) leverage vast datasets of weakly aligned image-text pairs to strengthen the connection between vision and language tasks. While scaling laws in pretraining have been well studied, the impact of scaling laws on downstream vision tasks has been less explored. Understanding these dynamics is critical for optimizing model design and performance in downstream vision scenarios.

### 6.2 SCALING LAW IN DOWNSTREAM TASKS

Beyond upstream pretraining, significant attention has been directed towards scaling laws in downstream tasks. Studies like (Liu et al., 2024; Xia & Huang, 2024) investigate neural scaling laws on graph-based models from both model and data perspectives. SMLPer-X (Cai et al., 2024) constructs a large-scale human pose and shape estimation dataset, creating a foundational model. Other studies, like (Minderer et al., 2024; Tschannen et al., 2024) focus on expanding training data size. However, these works often attempt to address isolated scaling aspects without establishing a universal scaling law for downstream vision tasks. In this work, we aim to address this gap by investigating general scaling laws in downstream vision tasks.

## 7 CONCLUSIONS

In this work, we explore the scaling law in downstream vision tasks. Firstly, we examine the three key factors of scaling laws: model size, data volume and input resolution, discovering similar trends to pretraining tasks. To address the optimization challenges in naive training, we introduce the DT-Training approach. Additionally, we propose a closed-loop scaling strategy to iteratively enhance model performance. Our model surpasses existing counterparts on the GTrack Bench. Our approach can also be applied to other tasks such as object detection. These results highlight the effectiveness and generalization capabilities of our method.

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

Table 9: **Statics of current benchmarks.** Trajectories in current popular benchmarks are limited.

| | LaSOT (Fan et al., 2019) | LaSOT$_{ext}$ (Fan et al., 2019) | TrackingNet (Muller et al., 2018) | TNL2K (Wang et al., 2021c) | UAV123 (Mueller et al., 2016) | Sum |
|---|---|---|---|---|---|---|
| Trajectories | 280 | 150 | 511 | 600 | 123 | 1664 |
| Videos | 280 | 150 | 511 | 600 | 123 | 1664 |
| Mean Frames | 2512 | 2395 | 441 | 697 | 1247 | - |

Table 10: **Statics of training data.** We combine multiple datasets to create a large scale training data to conduct scaling up experiments.

| Datasets / Statics | LaSOT | GOT-10K | TrackingNet | COCO | TNL2K | UAVDT | MOT16 | MOT17 | MOT20 | DanceTrack | SportsMOT | TAO | UVO | MOSE | OVIS |
|---|---|---|---|---|---|---|---|---|---|---|---|---|---|---|---|
| Trajectories | 1400 | 10000 | 30600 | 118288 | 1300 | 2593 | 731 | 2388 | 2332 | 419 | 639 | 15997 | 95308 | 3210 | 2482 |
| Videos | 1400 | 10000 | 30600 | - | 1300 | 50 | 7 | 21 | 2 | 40 | 45 | 2921 | 6850 | 1307 | 407 |
| Mean Frames | 2512 | 156 | 472 | - | 560 | 814 | 759 | 759 | 2333 | 1044 | 635 | 1055 | 89 | 61 | 65 |

# A APPENDIX

## A.1 MORE RELATED WORKS

**Visual Object Tracking.** Visual object tracking aims to locate a target object in each frame based on its initial appearance. Traditional tracking methods (Bertinetto et al., 2016; Li et al., 2018; Zhang et al., 2020; Bhat et al., 2019; Danelljan et al., 2019; Li et al., 2019a; Bolme et al., 2010; Henriques et al., 2014; Chen et al., 2021; Yan et al., 2021a) use a two-stream pipeline to separate feature extraction from relation modeling. Recently, the one-stream pipeline have taken a dominant role (Ye et al., 2022; Cui et al., 2022; 2024; Bai et al., 2023; Wei et al., 2023; Chen et al., 2022; 2023; Gao et al., 2023) combining these processes into a unified approach. These one-stream models are primarily built on the vision transformer architecture, which utilizes a series of transformer encoder layers. This design enables more effective relationship modeling between the template and search frame, leading to impressive performance. While previous works enhance model performance by increasing model parameters or input resolution, they often rely on limited training data and have not systematically explored the scaling law in visual object tracking tasks.

## A.2 GTRACK BENCH

Existing tracking models (Cui et al., 2022; 2024; Ye et al., 2022; Bai et al., 2023) tend to evaluate performance on a limited set of benchmarks (about 3-4), as detailed in Table 9. These benchmarks offer limited trajectories and fall short of comprehensively evaluating a model's tracking capabilities. Thus we introduce the GTrack Bench, which consists of 12 challenging benchmarks. Among the 12 benchmarks, 10 are singel object tracking benchmarks, including LaSOT (Fan et al., 2019), LaSOT$_{ext}$ (Fan et al., 2019), TrackingNet (Muller et al., 2018), TNL2K (Wang et al., 2021c), UAV123 (Mueller et al., 2016), Avist (Noman et al., 2022), LaGOT (Mayer et al., 2024), La-TOT (Zhu et al., 2023b), HOOT (Sahin & Itti, 2023), and VideoCube (Hu et al., 2022). Additionally, it includes two datasets from VOS and VIS tasks, MOSE (Ding et al., 2023) and OVIS (Qi et al., 2022). These datasets emphasize real and complex scenarios, offering more challenging videos. By integrating these datasets, we construct a comprehensive evaluation suite with three times the number of trajectories (4369 in total), allowing for a more thorough assessment of model capabilities in real-world scenarios.

## A.3 TRAINING DATA

Currently, state-of-the-art tracking models (Cui et al., 2022; 2024; Ye et al., 2022; Bai et al., 2023; Wei et al., 2023) are trained on a combination of several datasets, including TrackingNet (Muller et al., 2018), LaSOT (Fan et al., 2019), GOT-10K (Huang et al., 2019), and COCO (Lin et al., 2014). However, these datasets alone are insufficient for fully training highly capable tracking models. We datasets from related tasks into a single object tracking format to create a large-scale training set. These datasets originate from tasks such as single object tracking (LaSOT (Fan et al., 2019), GOT-10K (Huang et al., 2019), TrackingNet (Muller et al., 2018), COCO (Lin et al., 2014), TNL2K (Wang et al., 2021c), and UAVDT (Du et al., 2018)), multi-object tracking (MOT16 (Milan et al., 2016), MOT17 (Dendorfer et al., 2021), MOT20 (Dendorfer, 2020), DanceTrack (Sun et al.,

2022), SportsMOT (Cui et al., 2023)), video object segmentation (MOSE (Ding et al., 2023)), video instance segmentation (OVIS (Qi et al., 2022)), and open-world object tracking and segmentation (TAO (Achal et al., 2020) and UVO (Wang et al., 2021a)). Statistics of these datasets are displayed in Table 10. By incorporating a substantial number of training trajectories, we expand our dataset to four times its original size, exceeding the capacity of the initial datasets. We conduct our scaling up experiments based on this large scale dataset.

