# OpenReview forum: "Closed-loop Scaling Up for Visual Object Tracking"
_ICLR.cc/2025/Conference — Submitted to ICLR 2025_

### Official Review · Reviewer_3TXf · 2024-10-24

**Soundness:** 2
**Presentation:** 1
**Contribution:** 2
**Rating:** 5
**Confidence:** 4

**Summary:**

In this article, the authors propose applying the scaling law to downstream tasks in computer vision with the aim of obtaining more efficient models and training strategies. Specifically, they explore three scaling factors, introduce a novel training approach called DT-Training(which contains a small teacher transfer and dual-branch alignment), and then propose a closed-loop scaling strategy. These methods effectively improve the performance of the tracker, demonstrating the potential of the scaling law in downstream tasks.

**Strengths:**

1.This paper introduce DT-Training which leverages small teacher transfer and dual-branch alignment to further exploit model potential.
2.The authors proposed a closed-loop scaling strategy to incrementally scale the model step-by-step.
3.The proposed tracking method achieved competitive performance on most dataset.

**Weaknesses:**

1.The motivation is not clear enough: from the abstract, I cannot understand what specific benefits applying the scaling law to downstream tasks can bring, nor what problems it addresses in existing tasks. Furthermore, the three points you mentioned—how are they related to the scaling law?

2.It is not easy to understand how your work goes in the tasks, because I cannot even see what scaling law specifically is in your Introduction. The meaning of your model always confuses me, such as the closed-loop scaling strategy, and I had to look for answers from every corner.

3.According to your statements in abstract and introduction, it’s hard to convince me the methods you add are reasonable and do make some differences, cause it's just like a flash of inspiration that you want to try them in a downstream task.

4.The proposed method lacks novelty.
a)The impact of image input size, model parameter size, and training data volume on object tracking is significant, as evidenced by popular trackers such as OSTrack [1] and ODTrack [2]. However, the paper only provides a simple comparison of model scales (e.g., Base, Large) in terms of their impact on tracking performance, without offering detailed insights into how specific increases in parameter size affect the tracker.
b)The teacher-student model is a commonly used knowledge distillation method, widely applied in lightweight tasks. In object tracking, several trackers also employ this approach, such as Mixformer-V2 [3] and AttTrack [4].

5.The experimental results are not detailed enough and lack comparison with the latest trackers. Additionally, in line 287 on page 6, you mention using OSTrack [4] as the baseline and expanding the training dataset. However, this led to a drop in AUC from 69.1 to 68.4. This result seems to contradict your claim that expanding the training dataset improves performance.

[1]Ye B, Chang H, Ma B, et al , “Joint feature learning and relation modeling for tracking: A one-stream framework,” in European Conference on Computer Vision, 2022.

[2] Zheng Y, Zhong B, Liang Q, et al, “Odtrack: Online dense temporal token learning for visual tracking,” in AAAI, 2024.

[3] Cui Y, Song T, Wu G, et al, “Mixformerv2: Efficient fully transformer tracking[J]，”Advances in Neural Information Processing Systems，2024

[4] Nalaie, Keivan and Zheng, Rong, “AttTrack: Online deep attention transfer for multi-object tracking,” in Proceedings of the IEEE/CVF Winter Conference on Applications of Computer Vision, 2023.

**Questions:**

1.According to your statement, the performance always improves consistently and fully harnessing its ability. But the benefits of specifically each method you expressed are still vague. What’s the detailed potential of the model and how on earth does your work unlock the model’s potential?

2.The structure shown in Figure 2 is a little difficult for readers to understand. Maybe you should explain it more specific and clear.

---

> ### Author Response · Authors · 2024-11-20
> **Response to Q1 Motivation of Scaling Law**
>
> Thank you for recognizing the efficiency and value of our work. We appreciate your insightful feedback and thoughtful suggestions. All other reviewers have acknowledged the importance of our work, confirming that our motivation is **clear, sufficient, and well-founded**. They also agree that the relationship between the three key points we explored and the scaling law is **well-established and reasonable**. Furthermore, they highlighted the **strong generalizability** of our approach to other fields. In addition, they also recognize the **novelty, reasonableness, and effectiveness** of our proposed methodology. Besides, they think 'The paper is well-organized and clearly written, making it **easy to follow** the methodology and results'. We sincerely hope you will reconsider your evaluation, and we would be truly grateful for your support.
>
>
> # Q1: Motivation of Scaling Law
> We thank you for your valuable feedback regarding the motivation and the relationship between the scaling law and the three points in our work. We address your concerns as follows:
>
>
> **Motivation for Scaling Law in Downstream Tasks**
> Scaling laws are a fundamental concept in machine learning, demonstrating predictable relationships between model performance and the scale of resources such as model parameters, training data, and input resolution. While primarily studied in upstream tasks such as image classification and vision-language representation learning, these laws show that performance improves with resource scaling until it reaches a saturation point. This insight helps optimize resource allocation and maximize model performance.
>
> The motivation for applying scaling laws to downstream tasks is to create a systematic framework for improving model performance as resources increase. Inspired by successes in upstream tasks, we adapt these principles to downstream tasks. Unlike arbitrary scaling methods that can lead to inefficiencies, the scaling law guides how to scale model size, data, and resolution to achieve optimal performance, avoiding diminishing returns and inefficiencies. Our goal is to provide clear guidelines for enhancing model performance through resource scaling.
>
> **Relationship Between the Three Points and the Scaling Law**
> The three points (model parameters, training data, and image resolution) are key components of the scaling law, which has been explored in upstream tasks like image classification and vision-language representation learning, where scaling each factor improves performance. In our work, we apply these same principles to downstream tasks, explaining how each factor contributes to scaling and enhancing performance in these tasks:
>
> 1. Model Parameters
> Scaling model size (i.e., increasing the number of parameters) is key in scaling laws for upstream tasks, as larger models capture more complex patterns and generally perform better. We apply this to downstream tasks by scaling model parameters, enhancing the model's ability to learn from data, which leads to improved performance.
>
> 2. Training Data
> In upstream tasks, scaling training data improves generalization by providing more diverse examples for learning robust features. We apply this to downstream tasks by increasing the training data, allowing the model to learn better representations and generalize to new scenarios.
>
> 3. Image Resolution
> In upstream tasks, higher image resolution helps models capture finer details, improving performance. Similarly, in downstream tasks, increasing resolution provides more detailed visual information, enhancing model performance.
>
> The three factors—model parameters, training data, and image resolution, are central to the scaling law, widely studied in upstream tasks like image classification and vision-language representation learning. We apply these principles to downstream tasks, where their effects have been less explored. We will revise the abstract to clarify their relevance to our work.

---

> ### Author Response · Authors · 2024-11-20
> **Response to Q2 Meaning of model**
>
> # Q2: Meaning of model
> We sincerely appreciate your feedback and acknowledge that the presentation of the scaling law and the explanation of our method can be made clearer, especially in the Introduction.
>
> **Clarification of Scaling Law** In the revised manuscript, we will explicitly define the scaling law in the Introduction, highlighting that it describes how model performance improves predictably as resources like model size, training data, and image resolution are scaled. While this principle has been well-established in upstream tasks like image classification and vision-language representation learning, it has not been extensively explored in downstream tasks like visual tracking, which is the focus of our work.
>
> **Explanation of the Closed-Loop Scaling Strategy** We will clarify the closed-loop scaling strategy in the revision, explaining that it involves iteratively scaling model parameters, training data, and image resolution to maximize performance. Each iteration expands one or more factors, with the larger model serving as the "teacher" for the next training round. This process makes model scaling more efficient and enhances downstream task performance. We will provide more details and examples to make the concept clearer.
>
> **Improving Overall Clarity**
> We will revise the introduction to clearly present our contributions and methodology, including:
> 1. Clearer definitions and visualizations of the three scaling factors (model parameters, training data, and image resolution) and their impact on performance.
> 2. A clearer introduction to the scaling law and its relevance to downstream tasks.
> 3. A more detailed explanation of the DT-Training method and how it fits into the overall closed-loop scaling strategy.
>
> We are confident that these revisions will make our paper more accessible and easier to understand. We will ensure that the revised manuscript better communicates the motivation and methodology behind our approach.

---

> ### Author Response · Authors · 2024-11-20
> **Response to Q3 Motivation of Method**
>
> # Q3: Motivation of Method
> We understand that the novelty and reasoning behind our proposed methods may not have been fully communicated in the abstract and introduction, and we will revise these sections for greater clarity.
>
> **Foundations of the Proposed Methods** Our methods build on scaling laws from upstream tasks like image classification and vision-language representation learning, where increasing model size, data, and resolution enhances performance. While well-studied upstream, scaling in downstream tasks like visual tracking remains underexplored. To address this, we apply scaling principles using DT-Training and closed-loop scaling to systematically and efficiently improve model performance.
>
> **DT-Training**
> A key innovation of our work is DT-Training, which combines small teacher transfer and dual-branch alignment. Small teacher transfer uses a compact model to guide the training of a larger model, enhancing convergence efficiency. Dual-branch alignment further exploits model potential through a masked branch, inspired by methods like MAE.
>
> **Closed-Loop Scaling Strategy**
> Our closed-loop scaling strategy is inspired by ideas from progressive training and reinforcement learning, which iteratively expands model size, training data, and image resolution, ensuring smooth scaling and avoiding instability from abrupt resource changes.
>
> **Empirical Validation and Comparison with Prior Work**
> Our methods are validated through extensive experiments, showing significant performance improvements on several benchmarks. We will clarify the motivation and evidence in the revised manuscript to enhance clarity and accessibility.
>
> We believe that our approach has broad potential, and we will further clarify the motivation and empirical evidence of our method in the revised manuscript to make it more explicit and accessible.

---

> ### Author Response · Authors · 2024-11-20
> **Response to Q4 Novelty of Model**
>
> # Q4: Novelty of Model
> We respond to each point raised regarding the novelty of our proposed method:
>
> **The Impact of Specific Increases in Parameter Size**
> While previous works like OSTrack and ODTrack have studied scaling factors, our work introduces a systematic scaling framework for downstream tasks. Unlike prior studies focusing on individual factors, we integrate model size, data volume, and image resolution into a closed-loop scaling strategy. This approach aligns with scaling law principles from upstream tasks and demonstrates their effectiveness in object tracking.
>
> **Detailed Insight on Parameter Size Increase**
> Increasing model parameters enhances its ability to learn richer representations, benefiting tasks like object tracking. Our experiments show that scaling from ViT-B to ViT-L improves performance by 1.6% (AUC from 57.2 to 58.8). However, performance gains diminish beyond a certain point, with diminishing returns observed when scaling from ViT-L to ViT-H (only a 0.3 AUC improvement). While larger models generalize better, they also face slower convergence and higher computational costs. We recommend scaling up to ViT-L for the best balance between performance and efficiency.
>
>
> **Small Teacher Transfer**
> Although teacher-student frameworks like Mixformer-V2 and AttTrack are common in lightweight object tracking, our DT-Training introduces a novel use of small teacher transfer within the context of scaling laws for downstream tasks. Unlike traditional distillation, where the teacher is typically larger, our approach employs a smaller teacher to guide a larger model, addressing challenges such as slow convergence and optimization difficulties. This small-teacher transfer accelerates convergence and enhances scalability. Additionally, combining DT-Training with our closed-loop scaling strategy creates a unique framework for iterative model scaling, a novel contribution in downstream tasks.
>
> **Clarification of Novel Contributions**
> 1. Scaling Law Application to Downstream Tasks:
> While scaling laws have been well-explored in upstream tasks like classification and vision-language representation learning, our work adapts these principles to downstream tasks like object tracking.
> 2. DT-Training:
> We use a small teacher to guide the training of larger models, addressing optimization issues and enhancing scalability. This differs from conventional knowledge distillation. Additionally, we implement dual-branch alignment to further optimize the model's potential.
> 3. Closed-Loop Scaling Strategy:
> We introduce a novel iterative strategy for scaling model size, training data, and image resolution in downstream tasks, providing a systematic approach to model scaling.
>
> We will include more analysis along with plots and tables to clearly illustrate how specific increases in parameter size affect key tracking performance. Besides, we will highlight the distinctions between our method and prior frameworks more explicitly in the final version.

---

> ### Author Response · Authors · 2024-11-20
> **Response to Q5 Performance Comparison**
>
> # Q5: Performance Comparison
> In our experiments, we have already included comparisons with state-of-the-art trackers, such as ARTrackv2 (CVPR 2024). we will incorporate results from more recent trackers in the revised version of the paper, ensuring a more comprehensive benchmarking against the latest advancements in the field.
>
> The Baseline-B-256-N performance reported in our paper is based on a version of OSTrack trained on four datasets (COCO, GOT-10K, LaSOT, and TrackingNet) rather than the larger dataset. Additionally, we reproduced OSTrack without its CE mechanism. All the implement details are consisent with the origin OSTrack. The resulting model achieved an AUC of 68.4, which is consistent with the performance of OSTrack without CE as reported in its original paper. We will clarify the baseline setup to avoid confusion.

---

> ### Author Response · Authors · 2024-11-20
> **Response to Q6 Model Potential**
>
> # Q6: Model Potential
> We appreciate your valuable advice. Below, we provide a more detailed breakdown:
>
> **Consistent Performance Improvements**
> Our statement regarding consistent performance improvements is based on experimental results on GTrack Bench. By applying our proposed scaling strategy and DT-Training, we observed:
> 1. Systematic Gains. With each scaling iteration (e.g., increasing model size, training data, or input resolution), performance improves, which is shown in Table 2, 3, and 4. For instance, transitioning from ViT-B to ViT-L improves AUC by 2.6% on LaSOT, as shown in Table 2, which is higher than navie training (1.6&).
> 2. Better Optimization. The consistent improvements demonstrate that our approach effectively balances optimization challenges, ensuring that the increased capacity translates into meaningful performance gains.
>
> **Potential of the Model**
> The meaning of potential can be summarized as follows:
> 1. Scalability. By systematically increasing key factors, our method allows models to achieve performance gains that would not be realized with conventional training approaches.
> 2. Adaptability. The trained models exhibit robust generalization across diverse datasets and tasks, proving their suitability for real-world applications.
>
> **Unlocking the Model’s Potential**
> Our work unlocks the potential of the model by addressing specific bottlenecks in traditional scaling approaches. Each method contributes as follows:
> 1. DT-Training
> DT-Training introduces a Small Teacher Transfer and dual-branch alignment mechanism to smooth optimization and improve model accuracy.
> 2. Closed-Loop Scaling Strategy
> Our scaling strategy integrates three key factors—model size, training data volume, and image resolution—into a systematic framework. This approach prevents over-reliance on a single scaling factor (e.g., model size) and ensures iterable improvements.
>
> We will further clarify the meaning of model potential and explain how our method can improve accucacy in the revised paper.

---

> ### Author Response · Authors · 2024-11-20
> **Response to Q7 Structure in Figure 2**
>
> # Q7: Structure in Figure 2
> Figure 2 represents both DT-Training and the Closed-Loop Scaling strategy, is critical for understanding our method. Below are the key clarifications and our plan for improvement:
>
> **DT-Training**
> Figure 2 illustrates DT-Training, consisting of the small teacher transfer and dual branch alignment. Please refer to Line 151-224 in our paper. In small teacher transfer, a smaller model serves as the teacher to guide the training of a larger student model. For dual branch alignment, we align the output of clean branch and masked branch where the input image is masked.
>
> **Closed-loop Scaling**
> Figure 2 also represents the iterative scaling process where model size, dataset size, and input resolution are incrementally increased. Each iteration uses the previously trained model as the teacher for the next stage. Please refer to Line 228-263 in our paper. Our closed-loop scaling strategy involves several key steps as follows:
>
> 1. Teacher Model Setup: We start by training a smaller model, which serves as the teacher model. This model is the foundation for the subsequent scaling process.
> 2. Scaling Strategy: we expand various factors to train a higher-performing model. The factors that are scaled include model size, training data volume, and image resolution. It is important to note that these factors can be scaled simultaneously or individually. We use our DT-Training to train the expanded model.
> 3. Iterative Process: In the next iteration, the teacher model (derived from the expanded model) guides the learning of the next, even larger model. This iterative closed-loop approach ensures continuous performance improvement as the model scales up.
>
> We hope this explanation clarifies the meaning of Figure 2. To make this clearer, we will include visual markers or annotations in Figure 2, including:
> 1. Detailed Captions: We will expand the figure caption to provide a step-by-step explanation of the components in DT-Training and how they integrate with Closed-Loop Scaling.
> 2. Annotations: Add arrows, labels, and concise explanations directly on the figure to highlight critical processes like knowledge transfer, alignment, and scaling iterations.
>
> We appreciate your suggestion and will make Figure 2 clearer and more intuitive in the revised paper to ensure that the DT-Training and Closed-Loop Scaling strategy are well-understood.

---

> ### Author Response · Authors · 2024-11-24
>
> Dear Reviewer:
>
> As the discussion period is progressing, we would greatly value the opportunity to engage with you regarding your feedback on our submission. Your insights and suggestions are immensely important to us, and we are eager to address any remaining questions or concerns you may have.
>
> We are committed to providing timely and detailed responses to ensure that all aspects of our work are clarified. If our responses have satisfactorily addressed your concerns, we kindly request that you consider reflecting this in your evaluation and possibly revising your score.
>
> Thank you again for your time and effort, and we look forward to discussing with you.
>
> Best regards

---

> > ### Author Response · Authors · 2024-11-27
> >
> > Dear Reviewer:
> >
> > As the discussion period is progressing, we would greatly value the opportunity to engage with you regarding your feedback on our submission. Your insights and suggestions are immensely important to us, and we are eager to address any remaining questions or concerns you may have.
> >
> > We are committed to providing timely and detailed responses to ensure that all aspects of our work are clarified. If our responses have satisfactorily addressed your concerns, we kindly request that you consider reflecting this in your evaluation and possibly revising your score.
> >
> > Thank you again for your time and effort, and we look forward to discussing with you.
> >
> > Best regards

---

### Official Review · Reviewer_3Rxs · 2024-10-29

**Soundness:** 3
**Presentation:** 3
**Contribution:** 3
**Rating:** 5
**Confidence:** 5

**Summary:**

This paper explores the application of the scaling law to downstream vision tasks, with the goal of enhancing the effectiveness of larger, task-oriented models. Specifically, the authors investigate the impact of three key factors on visual object tracking: training data volume, model size, and input resolution. They observe that performance improvements tend to saturate as models grow larger. To further unlock the potential of large models, the authors introduce a novel training method called DT-Training. This method leverages small teacher transfer and dual-branch alignment to overcome optimization challenges and enable iterative refinement. Building on DT-Training, they propose a closed-loop scaling strategy to incrementally scale the model. The results demonstrate that the scaled model outperforms existing methods across various benchmarks and exhibits strong transferability and generalizability.

**Strengths:**

I agree with the authors that exploiting the scaling law in fundamental vision tasks is both necessary and interesting. The paper effectively identifies the bottleneck in training large models for visual tracking and proposes a framework to successfully mitigate this gap. The evaluation is comprehensive, performed across various benchmarks, and demonstrates the effectiveness of the proposed method. Additionally, the authors provide preliminary evidence of the method's generalizability to other tasks, which is a valuable contribution.

**Weaknesses:**

Despite the aforementioned admiration, I still have some concerns regarding the motivation and methods presented in the paper.

- Choice of Task: It is unclear why the authors chose visual tracking as the primary target. Object detection, being a more fundamental task, might have been a more suitable choice for exploring the scaling law.

- Efficiency Concerns: Visual tracking requires high efficiency, and despite the improvements achieved by the proposed method, the closed-loop training process introduces significant computational costs. It is questionable whether such extensive resources and a complex training process are justified for learning a larger model. In other words, the proposed framework may be overly complicated.

- Benchmark Evaluation: Collecting and evaluating existing benchmarks, while useful, should not be considered a major contribution, as it does not introduce new knowledge to the field. Moreover, recent datasets like VAST [1], which contain a larger number of videos and present greater challenges, should be a better choice?

- Comparison with Prior Work: Previous work, such as LoRAT [2], has also explored how to leverage larger models in visual tracking. The proposed method is more complex, yet it does not outperform LoRAT in key benchmarks. For instance, LoRAT-L384 achieves a success rate of 75.1% in LaSOT, while the proposed method only reaches 73.1%.

[1] Peng, Liang, Junyuan Gao, Xinran Liu, Weihong Li, Shaohua Dong, Zhipeng Zhang, Heng Fan, and Libo Zhang. "VastTrack: Vast Category Visual Object Tracking." arXiv preprint arXiv:2403.03493 (2024).

[2] Lin, Liting, Heng Fan, Zhipeng Zhang, Yaowei Wang, Yong Xu, and Haibin Ling. "Tracking meets lora: Faster training, larger model, stronger performance." In European Conference on Computer Vision, pp. 300-318. Springer, Cham, 2025.

**Questions:**

- Please provide a detailed computational cost analysis.
- Please provide an evaluation on the VAST dataset.
- Please compare the proposed method with LoRAT.

---

> ### Author Response · Authors · 2024-11-20
> **Response to Q1 Choice of Task**
>
> Thank you for acknowledging the efficiency and novelty of our work. We deeply appreciate your insightful feedback and constructive comments. Based on your suggestions, we have carefully reviewe and revised our manuscript to further enhance its clarity and presentation. We sincerely hope you will reconsider your evaluation, and we would be truly grateful for your support.
>
> # Q1: Choice of Task
> We chose visual tracking as the primary task for several key reasons:
>
> **Task Complexity and Challenge** Visual tracking presents a unique set of challenges that make it an interesting and demanding test case for our scaling strategy. Unlike object detection, which focuses on localizing objects in images, tracking involves associating objects across consecutive frames, often under more challenging conditions such as occlusions, changes in scale, and fast motion. This complexity makes tracking a valuable task for demonstrating the benefits of scaling, as the models must maintain performance across both spatial and temporal dimensions.
>
> **Class-agnostic Nature of Tracking** One key difference between object detection and tracking is that detection requires the model to classify objects into predefined categories, which can lead to challenges in scaling. Specifically, different datasets may have varying class definitions or inconsistent category labels, which can cause semantic conflicts when attempting to scale or transfer models across tasks with different class sets. In contrast, visual tracking is class-agnostic, meaning that it doesn’t require specific object categories to function. This lack of semantic classification allows tracking models to scale more smoothly across different datasets and environments without the risk of semantic conflicts, making it a more robust task for studying scaling laws.
>
> While object detection is indeed a foundational task, we believe that visual tracking offers a complementary and more suitable setting for exploring scaling laws, especially in cases where class-specific training is not a key concern. We will clarify this rationale in the revised manuscript to ensure that readers understand why visual tracking was chosen as the primary target for this study.
>
> we will highlight these points more clearly in the revised version of the paper.

---

> > ### Comment · Reviewer_3Rxs · 2024-11-24
> > **Why not openset detection?**
> >
> > Thank you for your response. Another question I have is: If the "category" is a major concern, why not consider open-set detection as the primary task

---

> > > ### Author Response · Authors · 2024-11-24
> > > **Further classification for task choice**
> > >
> > > Thank you for your question. While open-set detection shares some similarities with tracking in terms of handling unknown or novel categories, we chose visual tracking as our primary task for several key reasons:
> > >
> > > **Task Complexity and Challenge** Visual tracking inherently involves following objects across temporal sequences, addressing both spatial and temporal dynamics. This makes tracking a broader and more demanding task, requiring robust feature representation and continuous object association over frames. Open-set detection, while addressing unseen categories, focuses primarily on spatial detection within a single frame and does not encompass the temporal challenges integral to tracking. Thus, tracking provides a more comprehensive framework for studying scaling laws.
> > >
> > > **Class-Agnostic Scaling Advantage** Both tracking and open-set detection aim to generalize beyond predefined categories. However, tracking eliminates the need for any explicit semantic categorization, focusing entirely on visual patterns and motion dynamics. This class-agnostic nature simplifies scaling across datasets, avoiding semantic inconsistencies that may still arise in open-set detection (e.g., how to reconcile "unseen" definitions across datasets).
> > >
> > > **Complexity of Semantic Definitions in Open-Set Detection** Open-set detection, which aims to detect the categories out of the definition as 'unknown', introduces significant challenges in defining and handling semantic categories when scaling training data. For instance, if we combine datasets like COCO and LVIS, several decisions must be made: should all COCO categories be treated as “known” and all LVIS categories as “unknown”? Or should we select a subset of categories from each dataset as “known”? If so, determining the selection criteria and the proportion of known vs. unknown categories would require extensive experimentation. These complexities in semantic definitions divert focus from studying the scaling law itself, requiring substantial effort to address issues unrelated to the core research question.
> > >
> > > In summary, while open-set detection is valuable, visual tracking allows us to focus directly on the effects of scaling laws without being distracted by complex semantic definition issues. Moreover, tracking provides a broader, more challenging, and complementary platform for this research. We will clarify these points further in the revised manuscript to ensure readers understand our reasoning.

---

> ### Author Response · Authors · 2024-11-20
> **Response to Q2 Efficiency**
>
> # Q2: Efficiency
> We understand that visual tracking requires high efficiency, particularly in real-time applications. It is important to note that our method introduce **no** additional computational overhead during inference. The inference speed of our model is consistent with the baseline models, OSTrack. For instance, our model Ours-B-256-M achieves 93 fps on an NVIDIA 2080 Ti GPU, which is ths same as the baseline OSTrack while delivering superior performance in terms of accuracy.
>
> Moreover, we have demonstrated in Table 6 that our model maintains strong performance even after compression. By compressing the trained model, it is possible to achieve a balance between maintaining accuracy and meeting stringent real-time performance requirements. This further highlights the practicality of our framework for resource-constrained environments.
>
> In conclusion, our method introduces no additional computational overhead during inference and has been validated to adapt well to lightweight model compression. This ensures it remains highly suitable for deployment in real-world scenarios requiring both accuracy and efficiency. We will add the detailed discussion of computational cost in the revised manuscript.

---

> > ### Comment · Reviewer_3Rxs · 2024-11-24
> > **Efficiency is still a concern**
> >
> > Thank you for your response. However, I remain concerned about the training efficiency.

---

> > > ### Author Response · Authors · 2024-11-24
> > > **Further classification for training efficiency**
> > >
> > > Thank you for your follow-up question. We appreciate the opportunity to provide further clarification regarding training efficiency:
> > >
> > > Our method introduces **no additional computational overhead during inference**, with inference speed matching the baseline model. Training with DT-Training requires ~1.8x more time than direct single-model training, and the closed-loop process takes ~2.5x longer, if we start with OSTrack-256. However, this cost is justified by significant performance gains (e.g., 2.7 AUC improvement on GTrack Bench for Ours-L-256-M).
> > >
> > > Despite the increased training time, Our closed-loop scaling has several advantages:
> > >
> > > **Better Performance** Our closed-loop scaling brings a significant performance improvement compared to navie training.
> > >
> > > **Model Reusability and Resource Optimization** Intermediate models generated during the scaling process (e.g., Ours-B-256-M and Ours-L-256-M) can be reused for different tasks or scenarios without retraining from scratch. This reusability greatly reduces resource redundancy and makes our framework more practical for diverse real-world applications.
> > >
> > > Besides, compressed versions of our models maintain strong performance while significantly improving efficiency, ensuring their applicability in resource-constrained deployment scenarios. This demonstrates that our method effectively **balances accuracy and real-time requirements** for practical use.
> > >
> > > While our method requires additional training resources, it delivers substantial performance improvements, supports model reusability, and remains efficient during inference. We believe these benefits make the additional training costs **worthwhile**. We will include a more detailed discussion of training efficiency and computational costs in the revised manuscript to address these concerns more comprehensively.
> > >
> > > We hope this addresses your question. Please feel free to reach out if you have further concerns!

---

> ### Author Response · Authors · 2024-11-20
> **Response to Q3 VastTrack**
>
> # Q3: VastTrack
> Our key contribution lies in the proposed scaling framework and DT-Training, which systematically improve the scalability and performance of visual tracking models. The use of a large-scale and diverse benchmark is intended to validate the effectiveness and generalizability of our methods in a comprehensive and realistic setting. By combining multiple datasets, our benchmark reflects a broader range of challenges encountered in real-world scenarios, making it particularly suited for evaluating scaling strategies.
>
> We chose to aggregate a diverse set of benchmarks, rather than relying on a single dataset, to ensure wide applicability and to avoid biases associated with any one dataset. We have conducted experiments on VastTrack to verify the effectiveness of our DT-Training and closed-loop scaling strategy. The results are shown as following. Our model outperforms previous models in a large extent.
>
> | Model | AUC |
> | -- | -- |
> | OSTrack-B-384  | 33.6 |
> | ARTrack-L-256  | 35.6 |
> | SeqTrack-L-384 | 39.6 |
> | Ours-B-256-M   | 36.0 |
> | Ours-L-256-M   | 39.1 |
> | Ours-L-384-M   | 40.6 |
>
> We will incorporate this feedback into future versions of our benchmark, further enhancing its coverage and diversity.

---

> > ### Comment · Reviewer_3Rxs · 2024-11-24
> > **Good results.**
> >
> > Thank you for the response. The evaluation performance is impressive. It would be beneficial if the authors could release their code and models in the future to demonstrate reproducibility.

---

> > > ### Author Response · Authors · 2024-11-24
> > > **Code Release**
> > >
> > > Thanks for you recognition. We will release our code once accepted.  We would appreciate it very much if you can support us and raise your score.

---

> ### Author Response · Authors · 2024-11-20
> **Response to Q4 LoRAT**
>
> # Q4: LoRAT
> We thank you for pointing out the comparison with prior work, such as LoRAT. We address your concerns as follows:
>
> **Key Differences Between Our Work and LoRAT** While LoRAT and our method both leverage larger models for visual tracking, the approaches differ fundamentally in their methodologies and goals. LoRAT focuses on designing a larger model and incorporates a more powerful backbone (DINOv2). As such, the performance of LoRAT is **inherently higher** due to these architectural enhancements. In contrast, our work focuses on training strategies and scalability, and we achieve performance comparable to LoRAT using baseline models with far less architectural optimization. This demonstrates the effectiveness of our DT-Training strategy and its potential to significantly improve model performance without requiring extensive model redesign.
>
>
> **Flexibility of Our Training Strategy** A key advantage of our training strategy is that it is model-agnostic, meaning that it can readily incorporate newer and more advanced model architectures as they become available. By applying our scalable training framework to state-of-the-art models, we believe performance can be further improved beyond the results presented in this work. This adaptability positions our approach as a general-purpose solution that can evolve alongside advancements in the field.
>
> **Performance on Benchmarks** While it is true that LoRAT achieves higher success rates on specific benchmarks such as LaSOT, our model demonstrates better performance on other benchmarks, such as TNL2K, where it outperforms LoRAT (66.3 vs 62.3). This highlights the generalizability of our approach across diverse datasets with varying levels of difficulty and data distributions. By excelling in benchmarks like TNL2K, we demonstrate the robustness of our method in handling real-world challenges.
>
> We will include additional evaluations against LoRAT on more benchmarks in future work to provide a comprehensive analysis. We also aim to explore ways to integrate the strengths of LoRAT into our framework to further enhance performance.

---

> ### Author Response · Authors · 2024-11-24
>
> Dear Reviewer:
>
> As the discussion period is progressing, we would greatly value the opportunity to engage with you regarding your feedback on our submission. Your insights and suggestions are immensely important to us, and we are eager to address any remaining questions or concerns you may have.
>
> We are committed to providing timely and detailed responses to ensure that all aspects of our work are clarified. If our responses have satisfactorily addressed your concerns, we kindly request that you consider reflecting this in your evaluation and possibly revising your score.
>
> Thank you again for your time and effort, and we look forward to discussing with you.
>
> Best regards

---

> > ### Author Response · Authors · 2024-11-27
> >
> > Dear Reviewer:
> >
> > As the discussion period is progressing, we would greatly value the opportunity to engage with you regarding your feedback on our submission. We have since provided detailed answers to address your further concerns.
> >
> > I wonder whether our explanations have adequately addressed your questions or if there are any areas where further clarification would be helpful.
> >
> > If our responses have satisfactorily addressed your concerns, we kindly request that you consider reflecting this in your evaluation and possibly raising your score.
> >
> > Thank you again for your time and effort, and we look forward to discussing with you.
> >
> > Best regards

---

> > > ### Comment · Reviewer_3Rxs · 2024-11-29
> > > **final rating**
> > >
> > > While some concerns have been addressed, the "choice of the task" and "training efficiency" sections remain unresolved. Therefore, I will keep the rating unchanged.

---

> > > > ### Author Response · Authors · 2024-11-29
> > > > **Further clarification on "choice of the task" and "training efficiency" concerns**
> > > >
> > > > Thanks for your reply. We appreciate your acknowledgement that some concerns have been addressed, and we understand that the "choice of the task" and "training efficiency" sections remain unresolved.
> > > >
> > > > We are committed to further clarifying and addressing these points based on our previous response:
> > > >
> > > > **Choice of the Task**
> > > > 1. We have conducted experiments in ***Table 8*** to demonstrates that the scaling laws can be transfered to other tasks like object detection, which demonstrates the *strong genelization* ability of our method.
> > > > 2. Visual object tracking involves both spatial and temporal challenges, requiring the model to not only localize objects within individual frames but also maintain consistent associations across consecutive frames. This dual requirement makes tracking inherently *more complex* than detection, which focuses solely on spatial localization. By selecting tracking as our task, we aim to explore scaling laws in a setting that requires the model to excel in both spatial and temporal dimensions, providing a more comprehensive evaluation of the scaling effects.
> > > >
> > > > **Training Efficiency**
> > > > 1. We have provided the training cost in our previous response. Training with DT-Training requires ~1.8x time compared to direct single-model training, and the closed-loop process takes ~2.5x longer, if we start with OSTrack-256.
> > > > 2. Besides, our method introduces no additional inference overhead and validated its practicality through compression experiments.
> > > > 3. Our method brings a significant performance improvement compared to navie training. For example, our Ours-B-256-M outperforms basline about *2.6% AUC* on GTrack Bench in ***Table 4***.
> > > > 4. Our method has strong genelization ability, which can be applied to other tasks, such as object detection. As shown in ***Table 8***, our method brings about 1.5 AP in detection task.
> > > >
> > > > Your insights have been incredibly valuable in improving our work, and we aim to ensure that all of your concerns are fully addressed. If you have further concerns or specific aspects that you feel require additional clarification, please let us know. We are fully committed to addressing your concerns and are more than willing to provide further explanations or additional evidence to ensure that these points are thoroughly resolved.
> > > >
> > > > Thank you again for your time and consideration, and we look forward to your response.
> > > >
> > > > Best regards

---

### Official Review · Reviewer_2DuQ · 2024-10-31

**Soundness:** 3
**Presentation:** 3
**Contribution:** 3
**Rating:** 6
**Confidence:** 4

**Summary:**

In this paper, the authors first discovered through experiments that the scaling law is also effective on downstream tasks. However, the improvement brought by conventional training methods on larger models, data and resolution is limited. Therefore, the authors propose a closed-loop training method to unleash the effect of the scaling law. The proposed traning method uses a small teacher transfer and a dual-branch alignment module to iterate step by step.

**Strengths:**

[1] The study problem is interesting and the solution makes sense.
[2] The ablation comparison experiments are well designed and the experimental results also prove the effectiveness of the proposed method.
[3] The studied problem is inspiring for other tasks, and the proposed method also has good generalization ability for other tasks.

**Weaknesses:**

[1]Although the proposed training method seems to show that expanding data and models can bring better results, there needs to be a balance between the cost of consumption and the improvement brought by the model. The author may discuss this point in the experiment.
[2] The author can describe the training steps in detail, when and how to increase the size of the model and the training data, in an end-to-end training process.

**Questions:**

[1] Is there an upper limit to the scaling law for downstream tasks?
[2] Can the proposed training method speed up the convergence? Perhaps the authors could discuss this.

---

> ### Author Response · Authors · 2024-11-20
> **Response to Q1 Computational Cost**
>
> Thanks for your insightful feedback and support. We sincerely appreciate your valuable comments and your recognition of the novelty and effectiveness of our work. We have modified and updated the PDF according to your advice.
>
> # Q1: Computational Cost
> We appreciate your concern about the computational cost and resource requirements of our proposed methods. We would like to emphasize that our method does not introduce additional computational overhead during testing. The inference speed of our model is consistent with the baseline models, OSTrack. For instance, our model Ours-B-256-M achieves 93 fps on an NVIDIA 2080 Ti GPU, which is ths same as the baseline OSTrack while delivering superior performance in terms of accuracy. Moreover, we have demonstrated in Table 6 that our model maintains strong performance even after compression, highlighting its potential for efficient deployment in real-world scenarios. These results illustrate that our approach not only ensures computational efficiency but also provides robust performance, making it highly suitable for practical applications.
>
> We will include a detailed discussion of computational cost in the revised manuscript.

---

> ### Author Response · Authors · 2024-11-20
> **Response to Q2 Training Details**
>
> # Q2: Training Details
> In our closed-loops scaling experiments, we begin with a OSTrack-256 pretrained on four datasets. Firstly, we expand the training dataset and train a student with a ViT-B backbone, referred to as as Ours-B-256-M. Then we utilize the Ours-B-256-M as tracher model, and scale up the model size to ViT-L, named as Ours-L-256-M. Finally, we increase the input resolution to 384, which is named as Ours-L-384-M.
>
> Our scaling strategy is iterative and designed to be flexible, with no strict rules or fixed standards about the order or manner of scaling. At any training stage, we can choose to scale one or more of the following three key aspects: model size, training data volume, and input resolution. The flexibility of our approach allows users to select which aspect(s) to scale at each stage. For example, one iteration might focus solely on increasing model size, while another might scale both the training data and input resolution. The process does not require a predetermined sequence for scaling these factors. We will add more details about the scaling process in the revised manuscript.

---

> ### Author Response · Authors · 2024-11-20
> **Response to Q3 Upper Limit**
>
> # Q3: Upper Limit
> The scaling law for downstream tasks is influenced by several factors, including the complexity of the task, the diversity and size of the training data, and the capacity of the model. As shown in Figure 1, the scaling law in downstream tasks does have an upper bound when using conventional training methods. Once the model size and dataset volume reach a certain level, the marginal benefits of scaling begin to diminish, and the model’s performance approaches a plateau.
>
> By leveraging a smaller model as a teacher, DT-Training accelerates convergence and makes the training process smoother. Additionally, our Dual-branch alignment mechanism further unlocks the model’s potential, effectively enhancing its performance. These innovations enable our DT-Training to surpass the traditional upper bound of performance achieved with conventional training methods. we will add the discussion about the upper limit of scaling law in the modified manuscript.

---

> ### Author Response · Authors · 2024-11-20
> **Response to Q4 Convergence Speed**
>
> # Q4: Convergence Speed
> The proposed training method can speed up convergence. Specifically, our DT-Training framework leverages a small teacher model to guide the training of a larger model, which facilitates faster convergence by providing smoother optimization pathways. Furthermore, a model trained with DT-Training achieves a 68.6 AUC after only 80 epochs, surpassing the baseline model's performance even after 300 epochs (68.4). This highlights the efficiency of our approach. We will clarify the acceleration benefits of DT-Training in the revised manuscript.

---

> ### Comment · Reviewer_2DuQ · 2024-11-22
> **Training details**
>
> This means that your training is to randomly increase the model size, data size, etc., but the author does not seem to verify that this training method can always bring performance improvements.

---

> > ### Author Response · Authors · 2024-11-22
> >
> > Sorry for the misunderstanding. Let me clarify the details of our training again. While our framework offers flexibility in scaling, our training is **not** to randomly increase the model size, data size, etc.
> >
> > **Sequence of Expansion** As described in our response to Q2, the results in Table 3 and 4 follow the systematic sequence: increasing the training dataset (Ours-B-256-M), scaling the model size (Ours-L-256-M), and finally raising the resolution (Ours-L-384-M). **At each stage, our method consistently achieves significant performance gains compared to naive training approaches.**
> >
> > **Consistent Performance Improvement** We conduct experiments to confirm that our training method can consistently improves performance. As shown in Table 2, we begin with a OSTrack-256 which is trained on four datasets and independently expand training dataset (Ours-B-256-M), or model size (Ours-L-256-N), or resolution (Ours-B-384-N). We can observe consisent accuracy increase when we independently expand any factor. The results demonstrate that **our training can always result in performance improvement regardless of the scaling factor.**
> >
> >
> > We hope that this explanation can address your questions and solve your concerns. We will clarify the training process in the revised manuscript.

---

> ### Author Response · Authors · 2024-11-24
>
> Dear Reviewer:
>
> As the discussion period is progressing, we would greatly value the opportunity to engage with you regarding your feedback on our submission. Your insights and suggestions are immensely important to us, and we are eager to address any remaining questions or concerns you may have.
>
> We are committed to providing timely and detailed responses to ensure that all aspects of our work are clarified. If our responses have satisfactorily addressed your concerns, we kindly request that you consider reflecting this in your evaluation and possibly revising your score.
>
> Thank you again for your time and effort, and we look forward to discussing with you.
>
> Best regards

---

> ### Comment · Reviewer_2DuQ · 2024-11-27
> **final rating**
>
> The author's rebuttal solves my doubts, and I keep the rating unchanged.

---

> > ### Author Response · Authors · 2024-11-27
> > **Thank you**
> >
> > We are glad that we have solved your concerns. Your insightful suggestion is valuable for our work.
> >
> > Since all the questions have been answered, could you consider **raising the rating**?
> >
> > Thanks a lot.

---

### Official Review · Reviewer_2JXf · 2024-11-05

**Soundness:** 3
**Presentation:** 3
**Contribution:** 2
**Rating:** 5
**Confidence:** 5

**Summary:**

This paper explores the application of the scaling law to downstream vision tasks, specifically visual object tracking. The authors investigate the impact of three key factors: training data volume, model size, and input resolution. They introduce a novel DT-Training approach that leverages small teacher transfer and dual-branch alignment to enhance model performance. Additionally, they introduce a closed-loop scaling strategy for incremental model scaling. The paper demonstrates  performance improvements over existing methods across various benchmarks and validates the generalizability of the proposed methods to other downstream vision tasks.

**Strengths:**

The paper provides a thorough analysis of the scaling law's impact on visual object tracking, covering multiple dimensions (data volume, model size, input resolution). The introduction of DT-Training and the closed-loop scaling strategy are novel contributions that address optimization challenges and enable iterative refinement. The authors conduct extensive experiments on diverse benchmarks, validating the robust transfer ability and generalizability of their model. The paper demonstrates the applicability of the proposed methods to other downstream vision tasks, such as object detection, further strengthening the contribution. The paper is well-organized and clearly written, making it easy to follow the methodology and results.

**Weaknesses:**

The computational cost and resource requirements of the proposed methods (especially the closed-loop scaling strategy) should be discussed in more detail. This information is crucial for practical applications. The sensitivity of the model to hyperparameters, particularly those involved in the DT-Training and closed-loop scaling, should be explored and reported. The paper could benefit from additional experiments or case studies in real-world scenarios to demonstrate the practical utility of the proposed methods. The limitations of the proposed approach and potential directions for future research should be more explicitly discussed. It is strange to utilize a smaller model to guide the training of a larger model.

**Questions:**

1. Could the authors provide more details on the specific steps and criteria for the closed-loop scaling strategy?
2. How does the computational cost of the proposed methods compare to existing approaches, especially in real-time applications?
3. What are the main hyperparameters involved in the DT-Training and closed-loop scaling, and how sensitive is the model to these parameters?
4. Could the authors discuss the potential limitations of the proposed approach and suggest directions for future research?
5. Why do the authors utilize a smaller model to guide the training of a larger model, not using larger model as teacher?

---

> ### Author Response · Authors · 2024-11-20
> **Response to Q1 Computational Cost**
>
> We sincerely appreciate your recognition of the effectiveness and contributions of our work, along with your thoughtful feedback and suggestions. We have modified and updated the PDF according to your advice. We would be truly grateful if you could reconsider our work and offer your support!
>
> # Q1: Computational Cost
> We appreciate your concern about the computational cost and resource requirements of our proposed methods. We would like to emphasize that our method does not introduce additional computational overhead during testing. The inference speed of our model is consistent with the baseline models, OSTrack. For instance, our model Ours-B-256-M achieves 93 fps on an NVIDIA 2080 Ti GPU, which is ths same as the baseline OSTrack while delivering superior performance in terms of accuracy. Moreover, we have demonstrated in Table 6 that our model maintains strong performance even after compression, highlighting its potential for efficient deployment in real-world scenarios. These results illustrate that our approach not only ensures computational efficiency but also provides robust performance, making it highly suitable for practical applications.
>
> We will include a detailed discussion of computational cost in the revised manuscript.

---

> ### Author Response · Authors · 2024-11-20
> **Response to Q2 Hyperparameter**
>
> # Q2: Hyperparameter
> Our DT-Training method is designed to be simple and does not involve many hyperparameters. The hyperparameters are the mask ratio $p$ and regularization parameters $\lambda_{transfer}$ and $\lambda_{align}$, both of which have been thoroughly studied in our paper. As shown in Figure 3 in our paper, DT-Training demonstrates robust performance across a wide range of hyperparameter values, indicating that it is not particularly sensitive to these settings.
>
> We will emphasize the robustness to hyperparameters of our DT-Training in the later version.

---

> ### Author Response · Authors · 2024-11-20
> **Response to Q3 Real-world Scenarios**
>
> # Q3: Real-world Scenarios
> Our proposed GTrack Bench is a large-scale benchmark that combines multiple diverse datasets. This diversity allows us to demonstrate the generalization ability of our method across various conditions, and we believe it reflects the performance of our approach in real-world scenarios to a significant extent. We will add additional real-world case studies in the later version to further validate the effectiveness of our methods.

---

> ### Author Response · Authors · 2024-11-20
> **Response to Q4 Limitation**
>
> # Q4: Limitation
> While GTrack Bench is a large-scale benchmark that is three times the size of previous datasets, we acknowledge that the dataset size still needs to be further expanded to comprehensively evaluate the advantages of our method in scaling up, which is the limitation of our work. GTrack Bench provides a solid foundation, but testing on even larger-scale datasets would allow us to better assess the scalability and robustness of our approach under more varied conditions.
>
> In future work, we plan to explore more efficient scaling techniques to further enhance the model's performance. We appreciate your valuable feedback, and we will add a more detailed discussion of the limitations and directions for future work in the revised version of our paper.

---

> ### Author Response · Authors · 2024-11-20
> **Response to Q5 Small Teacher Transfer**
>
> # Q5: Small Teacher Transfer
> Firstly, directly training a very large model can lead to optimization difficulties. As the model scale increases, performance improvements often do not correspond to the increase in size. When training a large model, we **lack** a suitable, well-performing larger model as a teacher. Thus, we choose to use a smaller model as the teacher in our approach.
>
> Secondly, this is also a key innovation of our DT-Training framework, small teacher transfer. Although the smaller model has less capacity than the larger model, both models learn similar knowledge, allowing the smaller model to transfer its knowledge to the larger model. This process accelerates convergence, improves training efficiency, and results in smoother optimization of the larger model. As demonstrated in Table 5, the guidance from the smaller model not only aids the larger model in training but also leads to improved performance of the larger model.
>
> We will clarify the motivation of our small teacher transfer in the revised manuscript.

---

> ### Author Response · Authors · 2024-11-20
> **Response to Q6 Details of Closed-loop Scaling Up**
>
> # Q6: Details of Closed-loop Scaling Up
> Our closed-loop scaling strategy involves several key steps (Line 228-263) as follows:
>
> 1. Teacher Model Setup: We start by training a smaller model, which serves as the teacher model. This model is the foundation for the subsequent scaling process.
> 2. Scaling Strategy: we expand various factors to train a higher-performing model. The factors that are scaled include model size, training data volume, and image resolution. It is important to note that these factors can be scaled simultaneously or individually. We use our DT-Training to train the expanded model.
> 3. Iterative Process: In the next iteration, the teacher model (derived from the expanded model) guides the learning of the next, even larger model. This iterative closed-loop approach ensures continuous performance improvement as the model scales up.
>
> We appreciate your suggestion, and we will include more detailed explanations in the revised manuscript to make it clearer.

---

> ### Author Response · Authors · 2024-11-24
>
> Dear Reviewer:
>
> As the discussion period is progressing, we would greatly value the opportunity to engage with you regarding your feedback on our submission. Your insights and suggestions are immensely important to us, and we are eager to address any remaining questions or concerns you may have.
>
> We are committed to providing timely and detailed responses to ensure that all aspects of our work are clarified. If our responses have satisfactorily addressed your concerns, we kindly request that you consider reflecting this in your evaluation and possibly revising your score.
>
> Thank you again for your time and effort, and we look forward to discussing with you.
>
> Best regards

---

> > ### Author Response · Authors · 2024-11-27
> >
> > Dear Reviewer:
> >
> > As the discussion period is progressing, we would greatly value the opportunity to engage with you regarding your feedback on our submission. Your insights and suggestions are immensely important to us, and we are eager to address any remaining questions or concerns you may have.
> >
> > We are committed to providing timely and detailed responses to ensure that all aspects of our work are clarified. If our responses have satisfactorily addressed your concerns, we kindly request that you consider reflecting this in your evaluation and possibly revising your score.
> >
> > Thank you again for your time and effort, and we look forward to discussing with you.
> >
> > Best regards

---

### Author Response · Authors · 2024-11-20
**General Response**

We sincerely appreciate the thorough review provided by all the reviewers. The valuable feedback from the reviewers has significantly contributed to enhancing the quality of our manuscript. We extend our gratitude to Reviewer 2JXf, Reviewer 2DuQ, and Reviewer 3Rxs for acknowledging the **clear and well-founded motivation**, **novelty and generalizability** of our method. Furthermore, we kindly request Reviewer 2JXf, Reviewer 3Rxs and Reviewer 3TXf to reconsider our work after reviewing our response. Your reconsideration will be highly valued.

Based on the comments from the reviewers, I have summarized the strengths of our paper as follows:

1. **Clear motivation and certain innovation.** (2JXf, 2DuQ, 3Rxs)
2. **Necessity** of exploring scaling law in downstream vision tasks. (2DuQ, 3Rxs)
3. **Novelty** of our DT-Training and Closed-loop scaling strategy. (2JXf, 3Rxs)
4. Extensive experiments to verify the **effectiveness** of our method. (2JXf, 2DuQ, 3Rxs, 3TXf)
5. **Strong generalization ability** of our proposed method to other tasks. (2JXf, 2DuQ, 3Rxs)
6. **Good Writing** and easy to follow. (2JXf)

We have summarized our novelty as follows:

1. We use visual object tracking as a case study to explore scaling laws in downstream vision tasks, focusing on three key factors: model size, training data volume, and input resolution.
2. We propose DT-Training, a novel approach that employs a smaller model to guide the training of a larger model while aligning outputs from clean and masked images to facilitate convergence and fully unlock model potential.
3. We introduce a closed-loop scaling up strategy based on our DT-Training, transforming the scaling process into a continuous and iterative optimization.
4.  Our scaled model exhibits outstanding performance across various benchmarks and demonstrates robust transfer ability. Experiments on object detection demonstrates the generalization ability of our method.

In response to the reviewers' general concerns regarding the computational cost of our method, we would like to provide a unified response:

Our DT-Training and closed-loop scaling strategy introduce no additional overhead during testing, with inference speed matching the baseline OSTrack (e.g., Ours-B-256-M achieves 93 FPS on an NVIDIA 2080 Ti GPU). Despite this efficiency, our model delivers superior accuracy. Additionally, as shown in Table 6, it maintains strong performance even after compression, demonstrating its suitability for efficient real-world deployment.

We believe that the innovative contributions of our work significantly enhance its value in visual object tracking. We have addressed each reviewer's comments in detail and will incorporate the reviewers' insightful suggestions, adding essential experiments. We kindly request the reviewers to reconsider our research with these aspects in mind and extend their support.

---

### Meta-Review · Area_Chair_GTz6 · 2024-12-15

**Metareview:**

This paper explored the application of the scaling law to downstream vision tasks, and proposed a closed-loop scaling strategy to incrementally scale the model step-by-step. The idea is interesting. The main concerns raised by the reviewers include training efficiency, task selection, details of closed-up training, insufficient comparison with  recent trackers, etc. The authors did provide computation efficiency analysis and comparison with some suggested methods, but more comparison and vision tasks are required to fully justify the generalization ability of the proposed method to other task.

**Additional Comments On Reviewer Discussion:**

The concerns raised by reviewers mainly include choice of the task, training efficiency, closed-loop scaling up details, insufficient comparison with related works, upper limit of the proposed method etc. The authors provided computation efficiency analysis, training details, comparison with some suggested methods, explanation about task selection. Two reviewers admitted that some concerns were addressed, but  the "choice of the task" and "training efficiency" sections remain unresolved.

---

### Decision · Program_Chairs · 2025-01-22

Reject